



# A Methodology for the Spatiotemporal Identification of Compound Hazards: Wind and Precipitation Extremes in Great Britain (1979–2019)

Aloïs Tilloy[1], Bruce D. Malamud[1], Amélie Joly-Laugel[2]

[1]Department of Geography, King's College London, London WC2B 4BG, United Kingdom
[2]EDF Energy R&D UK Centre, Croydon CR0 2AJ, United Kingdom

*Correspondence to*: Aloïs Tilloy (alois.tilloy@kcl.ac.uk)

**Abstract.** Compound hazards are two different natural hazards that impact the same time period and spatial area. Compound hazards can have a footprint that can operates on different spatial and temporal scales than their component single hazards. This article proposes a definition of compound hazards in space and time and presents a methodology for the Spatiotemporal Identification of Compound Hazards (SI–CH). The approach is applied to the analysis of compound precipitation and wind extremes in Great Britain from which we create a database. Hourly precipitation and wind gust values for 1979–2019 are extracted from climate reanalysis (ERA5) within a region including Great Britain and the British channel. Extreme values (above the 99% quantile) of precipitation and wind gust are clustered with the Density-Based Spatial Clustering of Applications with Noise (DBSCAN) algorithm, creating clusters for precipitation and for wind gusts. Compound hazard clusters that correspond to the spatial overlap of single hazard clusters during the aggregated duration of the two hazards are then identified. Our ERA5 Hazard Clusters Database (given as a supplement) consists of 18,086 precipitation clusters, 6190 wind clusters, and 4555 compound hazard clusters. The methodology's ability to identify extreme precipitation and wind events is assessed with a catalogue of 157 significant events (96 extreme precipitation and 61 extreme wind events) that occurred in Great Britain over the period 1979–2019 (also given as a supplement). We find a good agreement between the SI–CH outputs and the catalogue with an overall hit rate (ratio between the number of joint events and the total number of events) of 93.7%. The spatial variation of hazard intensity within wind, precipitation and compound hazard clusters are then visualised and analysed. The study finds that the SI–CH approach (given as R code in supplement) can accurately identify single and compound hazard events and represent spatial and temporal properties of compound hazard events. We find that compound wind and precipitation extremes, despite occurring on smaller scales than single extremes, can occur on large scales in Great Britain with a decreasing spatial scale when the combined intensity of the hazards increases.

## 1 Introduction

The spatial and temporal scales of natural processes influence the spatial and temporal scales of the single or compound natural hazard that result (e.g., geomorphic: Naylor et al., 2017, Fan et al., 2019, Temme et al., 2020; atmospheric: Orlanski, 1975,



Mastrantonas et al., 2020 ; hydrologic: Blöschl and Sivapalan, 1995, Skøien et al., 2003, Diederen, 2019; ecologic: Schneider, 1994, Lancaster, 2018). Here, the spatial scale (the 'footprint') refers to the area over which the hazard occurs. The temporal scale is the duration over which the hazard acts on the natural environment. As displayed in **Fig. 1**, the extent of the temporal and spatial scales of natural hazards includes many orders of magnitude, influencing the relationship between natural hazards (Gill and Malamud, 2014; Leonard et al., 2014).

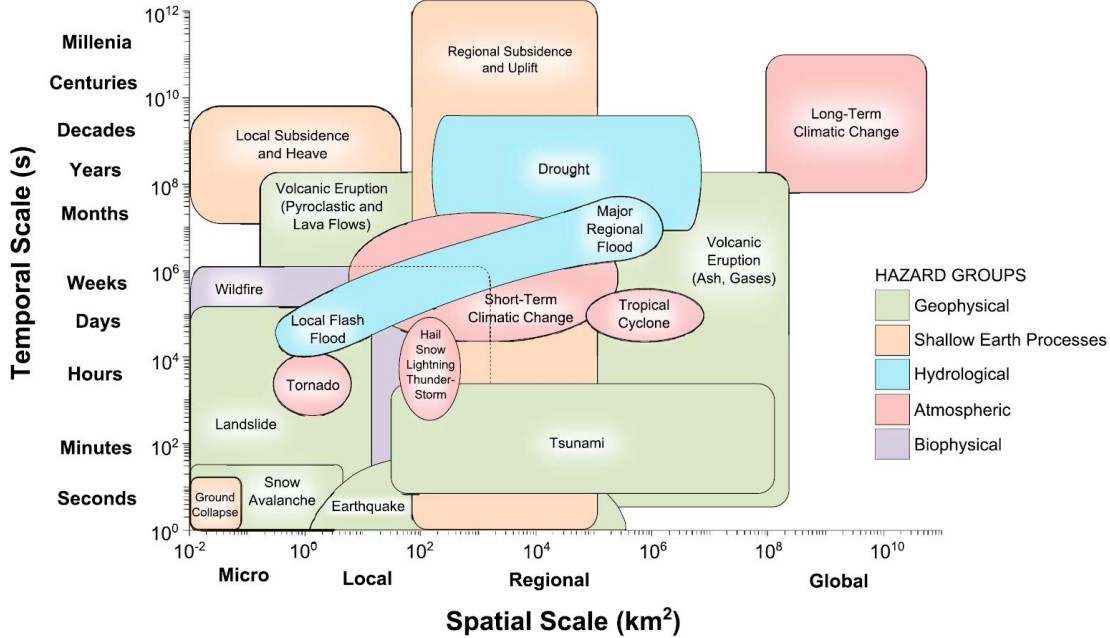

**Figure 1: Spatial and temporal scales of 16 natural hazards. Shown on logarithmic axes are the spatial and temporal scales over which the 16 natural hazards act. Here spatial scale refers to the area that the hazard impacts and temporal scale to the timescale that the single hazard acts upon the natural environment. Hazards are grouped into geophysical (green), hydrological (blue), shallow Earth processes (orange), atmospheric (red), and biophysical (purple). From Gill and Malamud (2014)**

Spatiotemporal clustering methods applied to environmental data can be powerful tools to understand the scales of different natural hazards by identifying natural hazard clusters (Barton et al., 2016). Such methods allow the extraction of spatiotemporal and intensity characteristics of natural hazard clusters. Estimating such characteristics is relevant when defining and understanding the potential impacts of natural hazards and their interrelation with society. Examples include the following:

- The duration of precipitation events (Yue, 2000; Vorogushyn et al., 2010) significantly affects dike failure, landslide
triggering, and flood losses.

- The increased duration of heatwaves significantly aggravates their health impacts (Nitschke et al., 2011).

- The spatial extent of drought influences its impact on society (García-Herrera et al., 2019; Balch et al., 2020)

In this article, we propose a robust methodology for the Spatiotemporal Identification of Compound Hazards (SI–CH), which we use to analyse the spatiotemporal features of wind and precipitation extremes in Great Britain at various scales (from hours

to days and from local to regional scale) during the period1979–2019. This SI–CH methodology is based on spatiotemporal





clustering of extreme values extracted from a gridded atmospheric dataset, the ERA5 climate reanalysis (Hersbach et al., 2019). Both extreme wind and precipitation are significant hazards in Great Britain (Pinto et al., 2012; Huntingford et al., 2014). These two hazards are usually associated with extratropical cyclones and severe storms (Zscheischler et al., 2020). Extreme wind and extreme precipitation have been defined as compound hazards (i.e., statistically dependent without causality) (Tilloy

et al., 2019).

In this article, we propose a robust methodology for the Spatiotemporal Identification of Compound Hazards (SI–CH), which we use to analyse the spatiotemporal features of wind and precipitation extremes in Great Britain at various scales (from hours to days and from local to regional scale) during the period1979–2019. This SI–CH methodology is based on spatiotemporal clustering of extreme values extracted from a gridded atmospheric dataset, the ERA5 climate reanalysis (Hersbach et al., 2019).

Both extreme wind and precipitation are significant hazards in Great Britain (Pinto et al., 2012; Huntingford et al., 2014). These two hazards are usually associated with extratropical cyclones and severe storms (Zscheischler et al., 2020). Extreme wind and extreme precipitation have been defined as compound hazards (i.e., statistically dependent without causality) (Tilloy et al., 2019).

Events including precipitation and wind extremes have been identified as multivariate compound events (co-occurrence of

multiple hazards in the same geographical region, causing an impact) (Zscheischler et al., 2020). The combination of wind and precipitation extremes can result in different and more significant impacts than the sum of the individual impacts due to extreme wind and extreme precipitation (e.g., e.g., the access to a flooded power plant due to heavy rain hindered by strong winds or road blocked by fallen trees ) (Martius et al., 2016). Previous studies have quantified the co-occurrences of extreme wind and extreme precipitation at large scales (Raveh-Rubin and Wernli, 2015; Martius et al., 2016; Ridder et al., 2020) by using climate

reanalysis data, thus providing a spatiotemporal framework to study multiple variables. To detect the occurrence of extreme wind and extreme precipitation events, Raveh-Rubin and Wernli (2015) averaged wind and precipitation anomalies spatially and temporally, while Martius et al. (2016) used a threshold approach to identify extreme occurrences of wind and precipitation.

This article is organised as follows. **Section 2** provides a brief background to spatiotemporal clustering. Then, in **Sect. 3**, the

spatiotemporal clustering algorithm used in the study and the gridded data retained for the analysis are introduced. This is followed by **Sect. 4**, where the SI–CH methodology steps for creating compound hazard clusters are presented, and a definition of compound hazard events in space and time is proposed. **Section 5** presents the results, where we assess the ability of the SI–CH methodology to identify hazard events, with the resultant natural hazard clusters confronted with a set of 157 major hazard events that impacted Great Britain, 1979–2019. Spatiotemporal and intensity properties of detected single and

compound hazard clusters are then analysed and discussed. Finally, in **Sect. 6**, we discuss the limitations of the SI–CH methodology and opportunities for its generalisation to other compound hazards.





## 2 Spatiotemporal clustering brief background

Clustering is broadly defined as any process of grouping data by their similarities (Ansari et al., 2020). It is fundamental to data analysis in various disciplines (e.g., biology, epidemiology, communication, criminology) (Xu and Tian, 2015). Clustering

can be considered spatially, temporally, or the two together. In addition, two clusters of different hazards can intersect in time and space, forming a compound hazard cluster. Two hypothetical examples of spatial clustering for two hazards and their intersections to become a compound hazard event (CHE) are shown in **Fig. 2**.

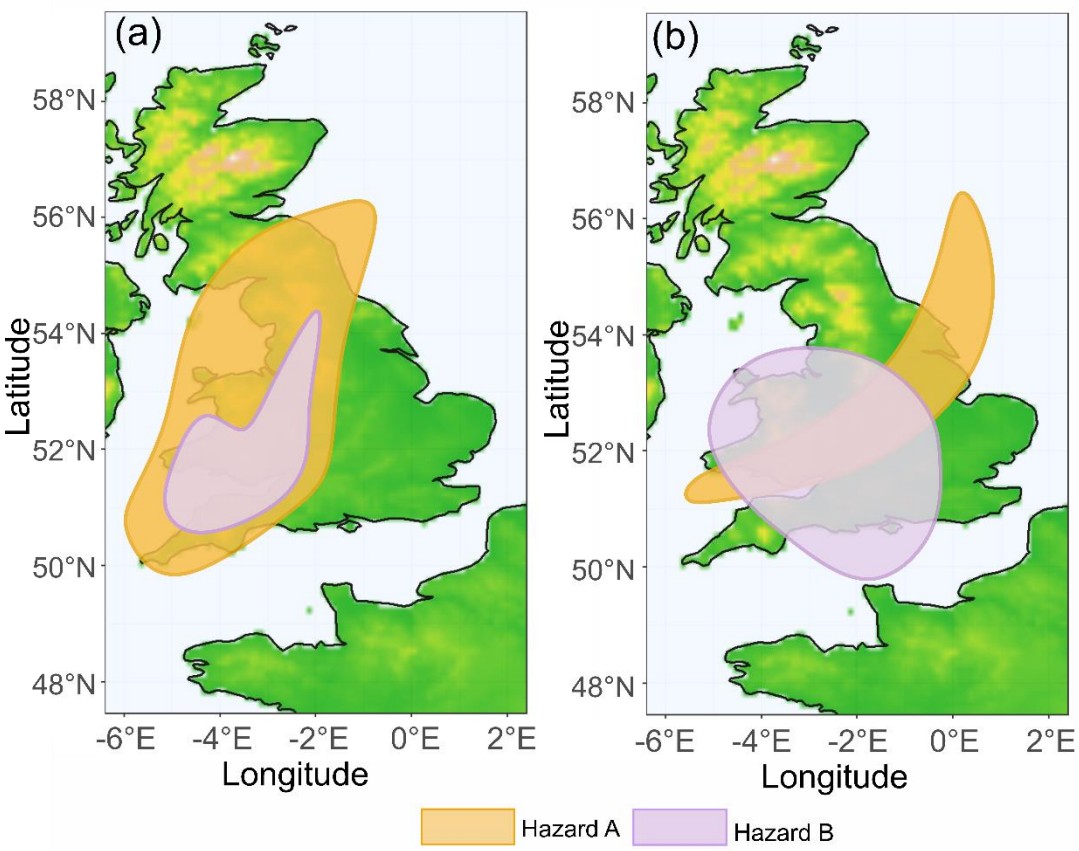

**Figure 2: Cartoon illustration of the spatial footprint of two hypothetical compound hazard events over Great Britain. Hazard A**
**(yellow) is a cluster of extreme occurrences of variable *x* and Hazard B (violet) is a cluster of extreme occurrences of variable *y*. In**
**(a) and (b) are shown two hypothetical examples of spatial overlaps, each a compound hazards event (CHE).**

**Fig. 2** illustrates compound hazard event spatially, but one can also examine compound hazard events overlapping in time, and both space and time together. The large increase in spatiotemporal data now available has created increased opportunities for spatiotemporal clustering approaches (Shi and Pun-Cheng, 2019; Ansari et al., 2020). Many methods have been developed

to cluster and classify data (e.g., partition, hierarchical, density-based, model-based clustering; see for a review Milligan and Cooper, 1987; Xu and Tian, 2015). Some clustering methods have been adapted to spatiotemporal clustering (e.g., Birant and



Kut, 2007; Agrawal et al., 2016; Yuan et al., 2017; Huang et al., 2019; Ansari et al., 2020). Spatiotemporal clustering is usually done on three different values characterising the data: two spatial coordinates and time (Ansari et al., 2020).

Three main approaches to spatiotemporal clustering exist (each with its own methods), including:

- *Point events clustering*: This approach aims to discover groups of events that are close to each other in space and time. It is used, for example, to cluster seismic events in time and space (Georgoulas et al., 2013).

- *Moving clusters*: This approach aims to detect behaviours of moving objects. While the identity of a moving cluster does not change over time, other attributes might change. An example is the spatiotemporal clustering of lightning strikes resulting from moving convective storms (Strauss et al., 2013).

- *Trajectory clustering*: This approach aims to capture groups of objects with similar movement behaviours, where the variable of interest is the movement itself (Yuan et al., 2017). Trajectory clustering contrasts with the moving cluster approach, where the moving object is of interest (vs. the movement itself). Examples of trajectory clustering include cyclone track clustering in different world regions (Ramsay et al., 2012; Rahman et al., 2018).

Among other factors, the characteristics of the data used influences the choice of the spatiotemporal clustering method. In this
article, we use climate reanalysis data, which are gridded data. Extreme occurrences of wind and precipitation are used to illustrate our methodology. We consider such occurrences as point events in time and space (see **Sect. 4.2**) and thus select a point events clustering approach.

## 3 Spatiotemporal data and study area

Spatiotemporal data includes information about the location (e.g., longitude and latitude) and time of the variable of interest
(Ansari et al., 2020). In this paper, we use climate data, focusing on extreme wind and extreme precipitation. However, the methodology we describe here can be applied to a wide range of variables. Spatiotemporal datasets of climate variables have been derived from many different sources, including the following: observations from instrumental stations and their interpolations (e.g., E-OBS, Cornes et al., 2018), climate model outputs/reanalysis (e.g., ERA5, Hashler et al.,2020) and remote sensing (e.g., CMORPH, Joyce et al., 2004). These have been used to analyse natural hazards and climate extremes in space
and time.

To ensure spatial and temporal consistency between wind and precipitation data, we extract both variables from a climate reanalysis dataset. Climate reanalysis offers homogeneous datasets for numerous environmental variables, including precipitation and wind gust, with different spatial and temporal resolutions. Those data are outputs of climate models calibrated on observed data worldwide (Brönnimann et al., 2018). Two primary climate reanalysis datasets include (i) the Climate
Forecast System Reanalysis (Saha et al., 2010) developed by the USA *National Centre for Atmospheric Research* and (ii) ERA5 (Hersbach et al., 2020) developed by the *European Centre for Medium-Range Weather Forecasts*. ERA5 (ECMWF Reanalysis 5th Generation) is used in the present study.





ERA5 was released in 2019 by ECMWF and benefits from the latest improvements in the field (Hersbach et al., 2020). The ERA5 data is available from 1979 to the present (we use 1979 up to September 2019), with a spatial resolution of 0.25° × 0.25° and an hourly temporal resolution. The data resolves the atmosphere using 137 levels from the Earth's surface up to a height of 80 km (Hersbach et al., 2020). ERA5 data are generated with a short (18 h) forecast of twice a day (06:00 and 18:00 UTC) and assimilated with observed data (Hersbach et al., 2020).

Reanalysis data are obtained from short-term model forecasts and can be affected by forecast errors; they are not observations (Pfahl and Wernli, 2012). Furthermore, reanalysis data offers a large amount of usable data for spatiotemporal clustering methods. This means that the methodology described in this study could be easily extended to other atmospheric or hydrological hazards (e.g., extreme temperature, Sutanto et al., 2020). The two following variables are extracted from the ERA5 product at a spatial resolution of 0.25° × 0.25°:

- *Extreme precipitation (p):* accumulated liquid and frozen water, comprising rain and snow, that falls to the Earth's surface in one hour (mm). This value is averaged over each 0.25° × 0.25° grid cell.

- *Extreme wind (w):* hourly maximum wind gust at a height of 10 m above the Earth's surface (m s$^{-1}$). The WMO (2021) defines a wind gust as the maximum wind speed averaged over 3 s intervals. As this duration is shorter than a model time step, this value is deduced from other parameters such as surface stress, surface friction, wind shear and stability. This value is averaged over each 0.25° × 0.25° grid cell.

When considering the study area boundaries, two factors should be considered: (i) the variability of climate, geology or topography within the study area. (ii) the possibility of not capturing an event in its totality because of edge effects (Cressie, 1993). Edge effects can potentially bias clustering analyses as edge points have fewer neighbouring cells than other cells within the domain (Cressie, 1993). To mitigate this issue, we set a buffer area of 2 cells at the edge of our study area (**Figure 3**). Clusters need to include extreme values (points) that are some distance away (here 2 cells) from the edge of the study area. A cluster of extreme values (points) exclusively within the buffer area will not be retained, but values in the buffer area can be part of other clusters. The study area chosen to illustrate the SI–CH methodology with extreme wind and precipitation contains most of Great Britain and part of northwest France (**Fig. 3**). The total area of the domain is 647,900 km$^2$, representing approximately a footprint of 500 km (33 cells) by 1200 km (45 cells), or a total of 1485 cells, each cell 0.25° × 0.25° (cells range from 18.6 km × 27.8 km in the south of the study region, to 14.3 km × 27.8 km in the north). The temporal resolution used is one hour over the period January 1979 to September 2019 (40 years and 9 months).



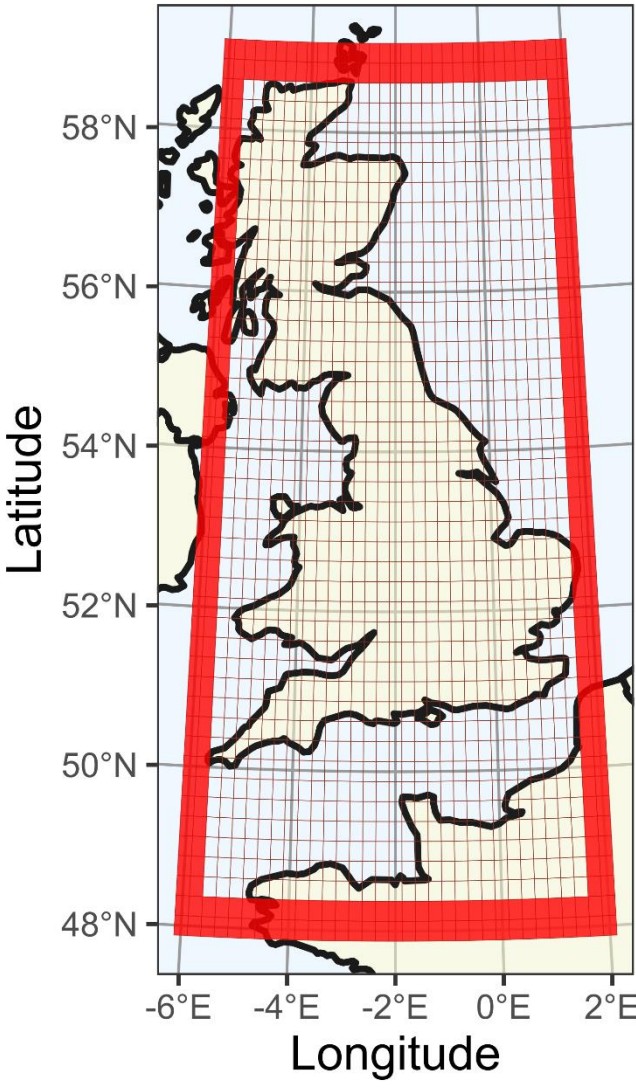


**Figure 3: Study area with a footprint of 1485 cells (45 cells × 33 cells, each 0.25° × 0.25°) used for spatiotemporal clustering of extreme precipitation and extreme wind for the period January 1979 to September 2019 at 1 h time steps using ERA5 data. The area includes most of Great Britain and parts of northwest France. The red frame is a buffer area of 2 cells (included in the total study area).**

Both Great Britain and northern France share the same temperate oceanic climate (Koppen climate classification Cfb) (Beck

et al., 2018). However, there are variations in precipitation and wind exposure within this broad region, particularly with

coastal areas being more exposed to high wind and mountainous areas being wetter (Hulme and Barrow, 1997). This variability

is accounted for in our methodology when sampling extreme events (discussed below in **Sect. 4**).

The use of a climate reanalysis product to study extreme events induces several limitations compared to observational data

(Donat et al., 2014; Angélil et al., 2016). In climate reanalysis, variables are computed over a grid cell, and the resulting value



is, therefore, an average. This often leads to a smoothing of local extreme values (Donat et al., 2014). The accuracy of reanalysis data also depends on various observation types (Hersbach et al., 2019). ERA5 benefits from the latest methodological improvements in data assimilation and modelling (Hersbach et al., 2018; ECMWF, 2020). Compared to its predecessor ERA-Interim, ERA5 offers finer spatial and temporal resolution, but most importantly, produces more accurate weather and climate

data in most regions of the world (Hersbach et al., 2019; Gleixner et al., 2020; Tarek et al., 2020). Despite these improvements, the spatial resolution is still relatively coarse and small scale convective events are still poorly captured as it is the case for most reanalysis products (Holley et al., 2014; Kendon et al., 2017; Beck et al., 2019). Furthermore, precipitation is not assimilated (calibrated on observations) in ERA5. Nevertheless, ERA5 seems to outperform other global reanalysis products for extreme precipitation (Mahto and Mishra, 2019) and captures most observed extreme precipitation events over Europe

(Rivoire et al., 2021).

## 4 Methodology: Spatiotemporal Identification of Compound Clusters (SI–CH)

We now discuss the methodology developed for spatiotemporal identification of compound hazard clusters using as illustration wind and precipitation extremes from ERA5 reanalysis (temporal resolution 1 h, January 1979 to September 2019; spatial resolution $0.25° \times 0.25°$) over Great Britain and northwest France. The specific clustering method used here to identify

spatiotemporal clusters of extreme wind and precipitation needs to comply with two characteristics of our spatiotemporal data:

(i) *The large size of the dataset:* ERA5 data is available for 40 years with an hourly timestep; this implies a significant amount of data over our study area of 1485 cells ($>5\times10^8$ values for each variable).

(ii) *Noise level:* The sample of extreme wind gusts and precipitation can produce extremes in individual and lone cells scattered in space and time, which cannot be associated with a specific hazard cluster.

To ensure flexibility in the specific point events clustering methodology developed, we do not assume a given shape for the natural hazard clusters. The characteristics of reanalysis climate data and the absence of assumptions about the shape of our hazard clusters guided the choice of a clustering algorithm toward the Density-Based Spatial Clustering of Applications with Noise (DBSCAN) algorithm (Ester et al., 1996; Hahsler et al., 2019). DBSCAN is a clustering algorithm for identifying clusters with arbitrary shapes (Shi and Pun-Cheng, 2019); see **Supplement 1** for further details. The DBSCAN algorithm is used here

as one part of our overall methodology to create spatiotemporal clusters of extreme wind and extreme precipitation. The methodology to create spatiotemporal clusters is described in **Fig. 4,** as a flowchart of the methodology steps:

- *Variable data extraction with thresholds.* Values of both variables (extreme wind, extreme precipitation) are extracted for the study area (**Fig. 4**). A threshold approach with a threshold $u$ is used to sample extreme values (discussed below, **Sect. 4.1**).

- *Single hazard spatiotemporal clusters.* The different parameters required for the clustering are set (see **Sect. 4.2**). Extreme values are clustered in space and time with a clustering algorithm (DBSCAN), creating two sets of clusters: (i) extreme wind and (ii) extreme precipitation.





- *Compounds hazard spatiotemporal clusters.* Extreme wind and extreme precipitation clusters are paired according to their spatiotemporal overlaps (see **Sect. 4.3**). The footprint of compound hazard clusters in time and space is then

identified, which allows the estimation of the spatiotemporal attributes.

The sensitivity of the procedure displayed in **Fig. 4** to the different input parameters is discussed and quantified in **Supplement 2**.

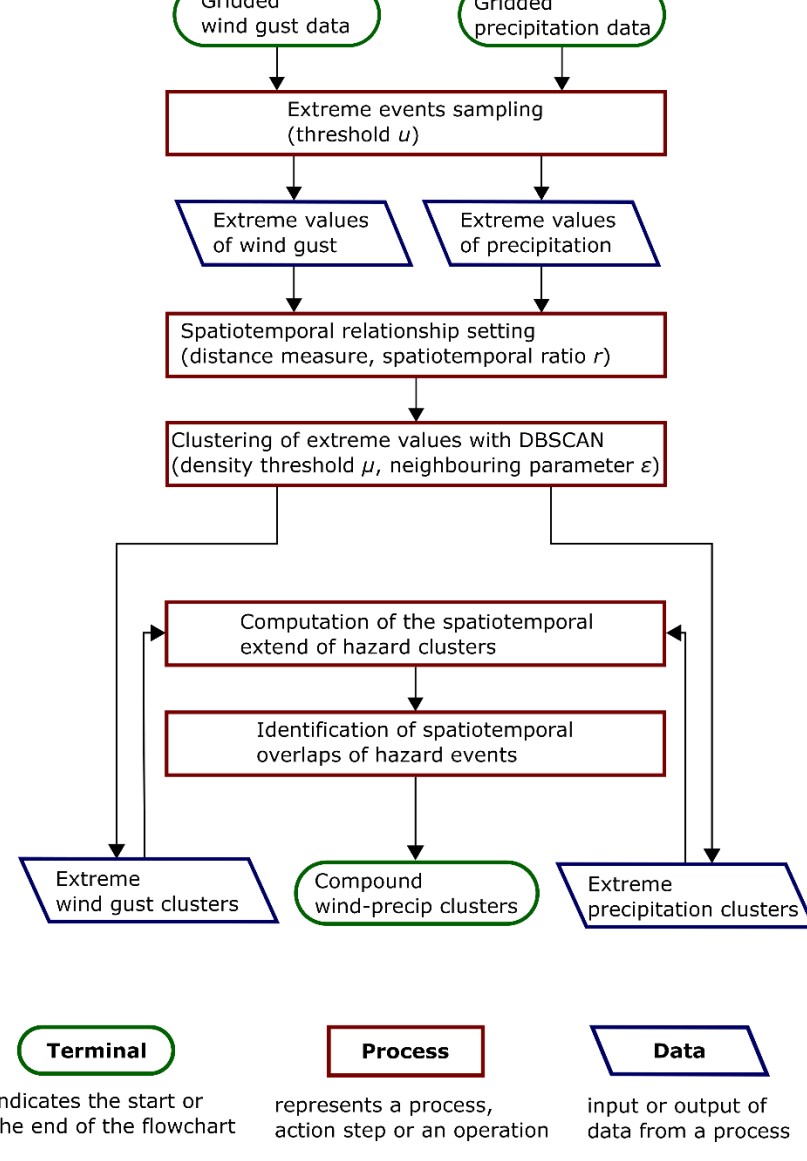

**Figure 4: Flowchart of the methodology developed, Spatiotemporal Identification of Compound Hazards (SI–CH), for wind and**
**precipitation data in Great Britain. DBSCAN (Density-Based Spatial Clustering of Applications with Noise) is an integral step in our methodology to identify compound hazard clusters in time and space. The name (terminal, process, data) and function associated with the three types of symbols used in the flowchart are given at the figure's bottom.**



## 4.1 Defining a hazard threshold

The methodology developed here uses the occurrences of climate variables above a given threshold to represent that climate
variable's extremes. These peaks over threshold serve as a proxy for the occurrence of natural hazards, in this case, extreme
wind and extreme precipitation. The use of a threshold to analyse the spatiotemporal occurrence of different extremes and their
potential combinations have been done on daily data by Martius et al. (2016), Sedlmeier et al. (2018) and Sutanto et al. (2020).
In the latter two studies, two approaches are used to define the value of a threshold: (i) an impact-based approach where the
threshold is related to a tipping point where impacts start occurring (Sedlmeier et al., 2018); (ii) a percentile-based approach
where the threshold is related to an empirical extreme quantile of the studied variable (Tencer et al., 2014; Visser-Quinn et al.,
2019; Sutanto et al., 2020). In the second approach, hazards are extreme events relative to the distribution of the studied
variable.

The percentile-based approach was chosen here as it provides a large sample size for robust statistical analysis. While not
being linked to a specific impact, the percentile-based approach can also be impact-relevant (Zhang et al., 2011) with extreme
occurrences of hourly maximum wind gusts and hourly accumulated precipitation, potentially negatively impacting society.
The connection between maximum wind speed and impact has been broadly acknowledged (Pinto et al., 2012). It has been
shown that a local 98th percentile is an impact-relevant wind threshold (Ulbrich et al., 2009). However, as our data are not
local, a 99th percentile was used to increase the probability of detecting potentially damage-relevant events. For the sake of
consistency, the same percentile is used for the definition of extreme events of both hazards. The threshold is computed for
each of the 1485 cells of the domain studied. The threshold values vary between $16.6 \leq w \leq 26.8$ m s$^{-1}$ for hourly maximum
wind gust $w$ and between $1.46 \leq p \leq 2.74$ mm h$^{-1}$ for precipitation $p$. The value of the selected percentile (here 99th) and the
corresponding threshold value significantly influence the clustering procedure (**Supplement 2**).

The threshold values $w$ for wind gust and $p$ for precipitation over the study area are displayed in **Fig. 4.** In this figure, the wind
gust threshold is higher in coastal regions as well as the north of England, Scotland and Wales. This contrast with south England
and northwest France which have significantly lower threshold values. For precipitation, one can observe a clear division
between the eastern and western part of Great Britain, with the western part having significantly higher threshold values. The
sample of extreme events is then composed of two distinct sets: (i) occurrences of extreme wind gust and (ii) occurrences of
extreme precipitation. These extreme events are then represented as point objects with coordinates in space (latitude and
longitude) and time (date). Here, both hazards are studied separately before being paired into compound hazard events. The
clustering algorithm is then applied to the points representing extreme wind and precipitation values.





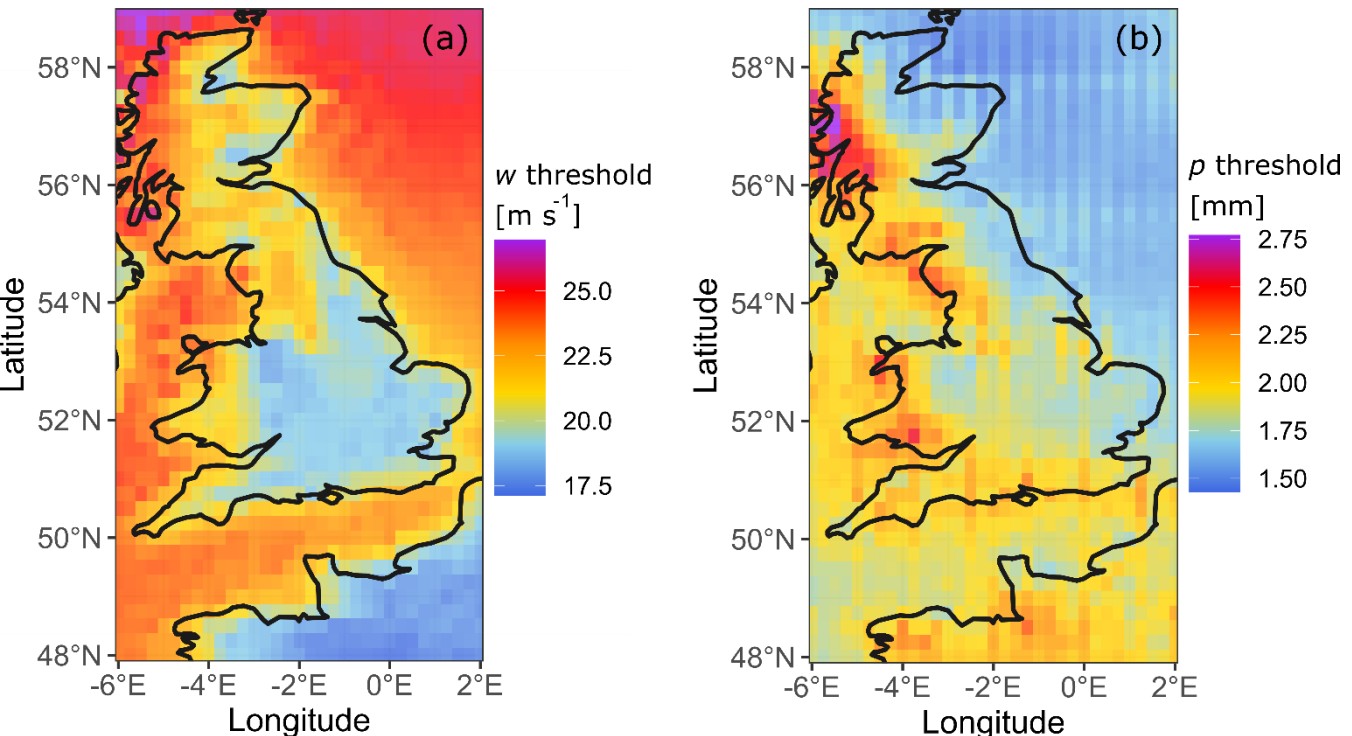

**Figure 5: Threshold values (see legends) used to extract extreme values for the clustering process over Great Britain and northwest France. The values correspond to the 99th percentile on each grid cell during the period 1979–2019 for (a) hourly maximum wind gust (w) and (b) hourly precipitation accumulation (p). Data from ERA5 (Hersbach et al., 2020).**

## 4.2 Construction of single hazard clusters

A method for sampling extreme values has been presented in **Sect. 4.1.** These extreme values are the input data for the construction of the cluster. In the present study, the spatiotemporal domain is assumed to be a space-time cube as done in other studies (e.g., Bach et al., 2014). The Euclidean distance is preferred to other distance measures in our study for simplicity. One of the advantages of this approach is that it is possible to take advantage of the spatial index structure (see **Supplement 1** for more details about the DBSCAN algorithm) to significantly speed up the runtime complexity (Hahsler et al., 2019). Three parameters inform the clustering procedure: (i) the relationship between spatial distance and temporal lag ($r$); (ii) the density threshold ($\mu$) for our cluster; (iii) the neighbour parameter ($\varepsilon$). These three parameters are now discussed:

### 4.2.1 First parameter: the spatiotemporal ratio $r$

The first step of our cluster event construction is to define the importance of spatial distance relative to temporal distance when computing the Euclidean distance between point objects. This is done according to physical considerations. Each points object in our input data represents one occurrence of an extreme event in one grid cell. Each grid cell is 0.25° latitude ($\simeq$27.8 km) by 0.25° longitude (ranging from 14.3 km in the southern part of our study area to 18.6 km in the northern part, **Fig. 2**). Grid cell areas ranging from 397 km$^2$ (in the south of our study area) to 517 km$^2$ (in the north). The temporal distance between each



extreme value is at least 1.0 h. Scaling factors are then introduced to give more importance to space or time distance in a three-
dimensional space-time cube (Ansari et al., 2020). We therefore express the spatiotemporal Euclidean distance $d_{p,q}$ (unitless)
between two point objects $p$ and $q$ as:

$$d_{p,q} = \sqrt{\left(ax_p - ax_q\right)^2 + \left(ay_p - ay_q\right)^2 + \left(bt_p - bt_q\right)^2} \tag{1}$$

with $x_p$ and $x_q$ the latitudes of the extreme value, $y_p$ and $y_q$ their longitudes, $t_p$ and $t_q$ their temporal coordinate, and $a$ and $b$ two
scaling parameters. The ratio $r = a/b$ is the spatiotemporal parameter controlling the relationship between spatial distance
and temporal lag. The scaling parameters are set to $a = 1/(0.25 \text{ deg}) = 4 \text{ deg}^{-1}$ and $b = 1 \text{ h}^{-1}$, giving a ratio $r = 4 \text{ h deg}^{-1}$. With
these parameter values, the three-dimensional space-time cube (**Fig. 6**) is normalised to be unitless, and each point object has
a spacing of 1.0 (unitless) in each dimension (longitude, latitude, time). In practice, this means that a distance of 0.25° in space
is weighted similarly to a distance of 1.0 h in time (**Fig. 6**). Nevertheless, even if each point is equally spaced in term of
longitude and latitude, this is not the case in term of geographical distance.

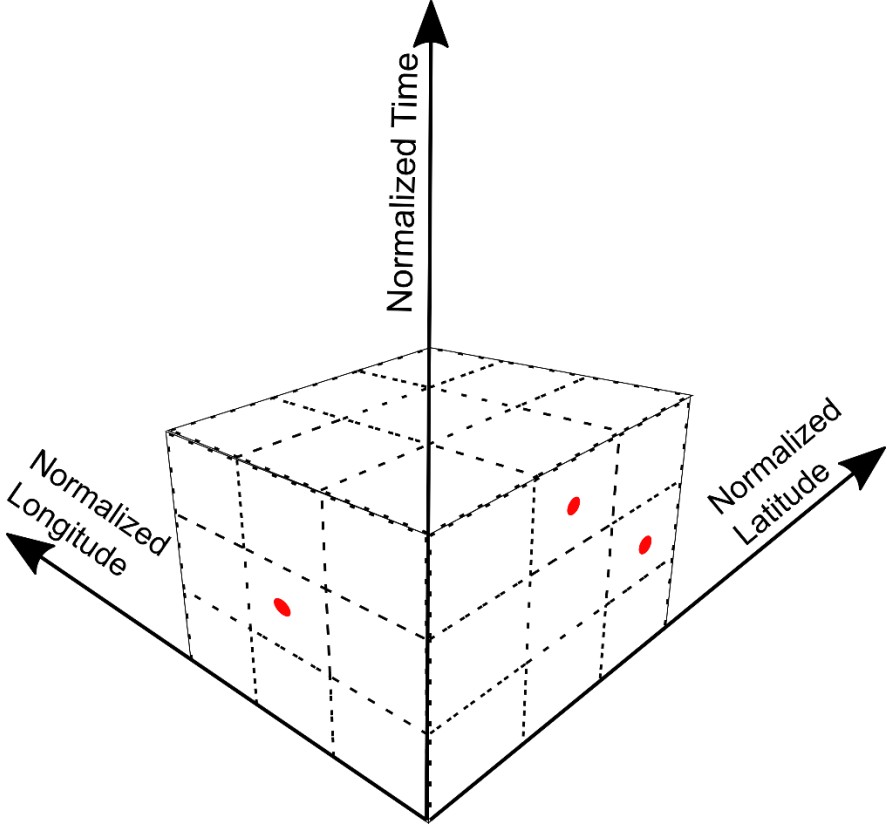

**Figure 6: Space-time cube as used in the SI–CH (Spatiotemporal Identification of Compound Hazards) methodology proposed in
this paper. The three small red dots represent extreme values. Each cube is of normalized latitude × normalized longitude ×
normalized time period. Each side of the cube is 1.0 and unitless, with normalization factors for latitude and longitude $a$ (in units of
deg$^{-1}$) and normalization factor for time $b$ (in units of time$^{-1}$). Normalized latitude and longitude for our ERA5 data is ($a \times 0.25$ deg),
with $a = 4$ deg$^{-1}$, and normalized time is ($b \times 1$ h), with $b = 1$ h$^{-1}$.**


### 4.2.2 Second parameter: the density threshold $\mu$

The density threshold parameter $\mu$ represents the number of neighbours a point needs to be considered a core point, and therefore generate a new cluster. This value needs to be greater than 4 points in our dataset (number of dimensions plus one). However, the detection of intense small scale events (e.g., the Bracknell storm, Berkshire, UK on 7 May 2000) is not intended here because of the relatively coarse resolution of ERA5 and its tendency to smooth local extremes. The aim is to detect different events of varying size. Small scale and short duration extreme precipitation and/or wind events in Great Britain are often associated with convective events. The size of such events varies from hundreds to tens of thousands of km$^2$ (Chazette et al., 2016; Rigo et al., 2019), while their duration goes from hours to days. Knowing that the area of the study area cells ranges between 400 and 520 km$^2$, we take a density threshold $\mu = 10$ points, meaning that events captured should last from at least 1 to 10 hours, cover a minimum area between 5200 and 400 km$^2$ while being composed of at least 10 extreme values.

### 4.2.3 Third parameter: neighbour parameter $\varepsilon$.

This third parameter is the neighbour radius $\varepsilon$ in which at least $\mu$ points (here $\mu = 10$) should be included to create a cluster. In this study, the neighbourhood is a spatiotemporal domain. This parameter controls the density of extreme events required to create a cluster. An optimal value for $\varepsilon$ depends on the dataset to be clustered and is assessed semi-automatically. The procedure to select a relevant radius for our wind and precipitation dataset is to plot the points' $k$–$NN$ distances (i.e., the distance to the $k^{th}$ nearest neighbour) in increasing order, to look for a knee in the plot. The distance to the $k^{th}$ nearest neighbour allows classifying data points by their similarity, here represented by their spatiotemporal distance. The idea behind this procedure is to separate points located inside clusters (with low $k$–$NN$ distance) from isolated noise points (with large $k$–$NN$ distance) (Hahsler et al., 2019). Here $k = \mu = 10$. More details about this step are available in **Supplement 1**.

### 4.2.4 Single hazard cluster parameters summary

The three parameters of the clustering procedure ($r$, $\mu$, $\varepsilon$) are now set. The spatiotemporal space has been discretised in a space-time cube (**Fig. 5**). Each grid point (representing one grid cell of input data) is spaced by a unit distance in each direction (longitude, latitude, time) with a unit distance representing 0.25° in the spatial dimension and 1.0 h in time. The density threshold ($\mu$) has been fixed at $\mu = 10$ points. A $k$ Nearest Neighbour ($k$–$NN$) search was performed, with $k = \mu = 10$ points. The result is a distance matrix containing the distance of each point to its 10–$NN$ allowing us to fix the neighbour parameter at $\varepsilon = 2.24$ for extreme wind and $\varepsilon = 2.45$ for extreme precipitation values. From this information, it is possible to estimate the spatiotemporal domain in which the 10–$NN$ needs to create a new cluster. This 10–$NN$ neighbourhood includes $n_{max}$=44 points with a maximum temporal distance of 2.0 h and a maximum spatial distance of 0.5° in latitude or longitude. The sensitivity of the clustering procedure to ($r$, $\mu$, $\varepsilon$) is assessed in **Supplement 2**.



### 4.3 Compound hazard events

One commonly used option to study compound extremes is to sample only the joint extreme events (i.e., extreme wind and extreme precipitation at a given location and time) (Martius et al., 2016; Tencer et al., 2016; Sutanto et al., 2020). However, when detecting the spatial and temporal characteristics of compound extremes, this option has the following weaknesses: (i) A high reliance on the spatial and temporal resolution of the input data in the definition of *compound*; (ii) Lack of considering the lag time between different extremes; (iii) Difficulty deciphering the spatial structure of extreme events. Our approach aims 305 to overcome these weaknesses.

Here, single hazard event clusters are created for both extreme wind and extreme precipitation. Compound hazard events are then detected by spotting the overlap of the extreme wind and extreme precipitation events in time and space. The footprint of a compound hazard event is the total area impacted for a duration of time. To define a compound hazard event's spatial and temporal scales, one can look at the overlap in time ($t$) and space ($S$) of single hazard event clusters. This overlap can be the 310 intersection AND ($t_{w \cap r}, S_{w \cap r}$) or the union OR ($t_{w \cup r}, S_{w \cup r}$) of the two hazard events in space and time. There are, therefore, four different possible definitions of a compound hazard event in space and time depending on the definition chosen for the overlap in space and time as displayed in **Figure 7**. The extent of the compound hazard event footprint widely varies depending on which combination of spatial and temporal overlap is retained. One can consider the following:

a) The duration of a compound hazard event can either be defined as the time during which both hazards occur (AND)
or as the aggregated duration of both hazards (OR). As the potential impact caused by a hazard can remain after the occurrence of this hazard (e.g., fallen trees blocking a road), the temporal scale of a compound hazard is then defined as the aggregated duration ($t_{w \cup r}$) of both single hazard events.

b) Footprints from different hazards need to overlap at least at one point to create a compound hazard event. The spatial scale of compound hazards is defined here as the intersection ($S_{w \cap r}$) of the spatial footprint of the two single hazards.

An overlap of the two hazards' footprint does not mean that the two hazards occur in the overlapping area at the same time (here same hour), but that the two hazards occurred, during at least one hour each, in that area during the same compound hazard event. This approach overcomes the weaknesses as mentioned above of constructing a joint occurrence sampling method without introducing a lag time (Klerk et al., 2015; Iordanidou et al., 2016). The time window in which a compound event can occur is flexible and fixed by the duration of both hazard events.



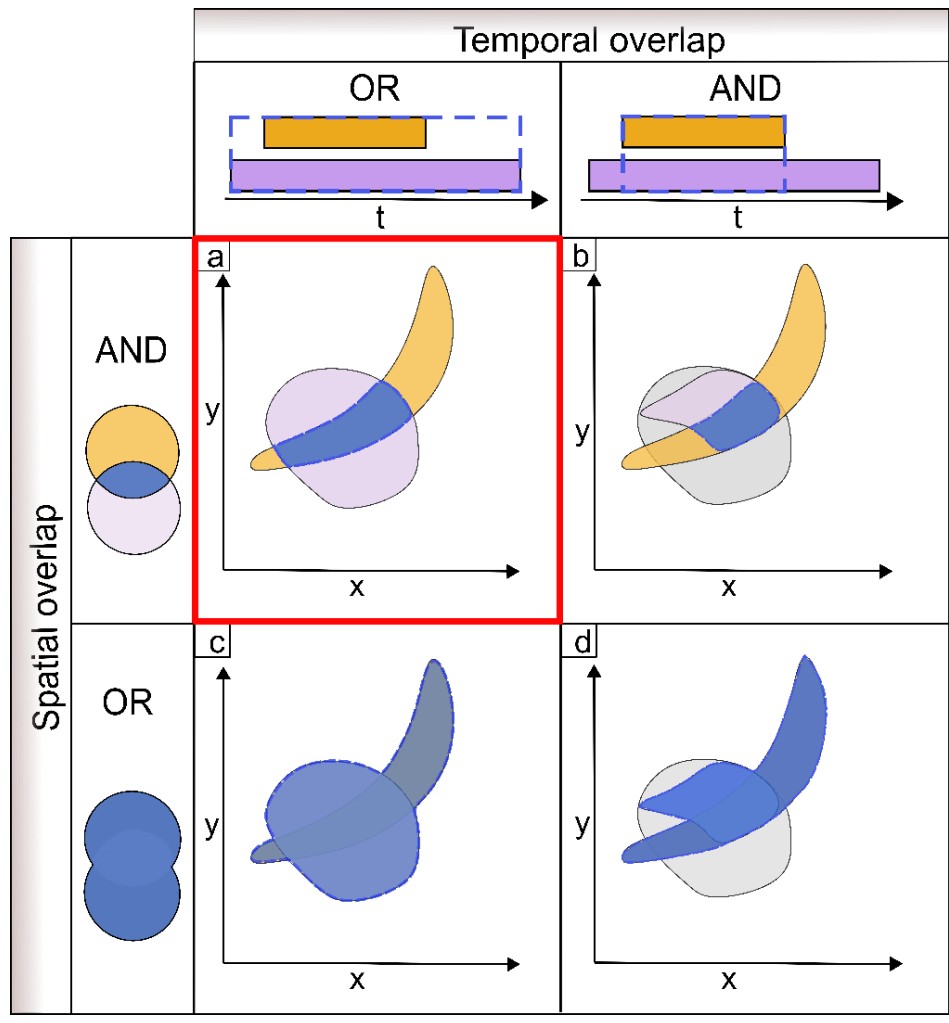

**Figure 7: Different spatial and temporal scales considered in this study to define compound hazard events, with each case representing a combination of spatial and temporal overlap. (a) [spatial AND] with [temporal OR], (b) [spatial AND] with [temporal AND], (c) [spatial OR] with [temporal OR], and (d) [spatial OR] with [temporal AND]. Hazard A is in orange, hazard B in purple, compound hazard in blue, and parts of footprints outside the temporal boundaries are in grey. The definition retained for the rest of the study is highlighted with a red frame (part a).**

We will use in our methodology a compound hazard event footprint (**Fig. 6a**) as the intersecting area (AND) on which two (or more) hazards develop during the aggregated union of the time periods (OR) of the two hazard events. From this definition and illustration in **Fig. 6a**, the spatial ($S$) and temporal ($t$) scales of a compound ("Comp") hazard event that includes wind ($w$) and precipitation ($p$) events are defined as follow:

$$t_{\text{Comp}} = t_{w \cup p} = t_w + t_p - t_{w \cap p} \; ; \quad S_{\text{Comp}} = S_{w \cap p} = S_w + S_p - S_{w \cup p}$$

$$(2)$$





with $t$ the duration and $S$ the area of the compound hazard event $(t_{\mathrm{Comp}}, S_{\mathrm{Comp}})$, wind event $(t_w, S_w)$ and precipitation event $(t_r, S_r)$. The duration of a compound hazard event corresponds to the union of the durations of both hazard events involved, meaning that $t_{\mathrm{Comp}} \geq \max(t_w, t_r)$. The paper examines compound wind–precipitation events; however, this definition aims to be applicable for other compound hazards (e.g., extreme hot temperature and drought).

**4.4 Single and compound hazard cluster attributes**

In our SI–CH methodology, each single and compound hazard cluster created is characterised by a set of attributes. Similarly to Visser-Quinn et al. (2019), three attributes (or metrics) are developed here: (i) intensity attributes, (ii) spatiotemporal attributes (iii) historical attributes as follows:

(i) *Intensity attributes for each and variable*.

a. Maximum precipitation accumulation ($p_a$).To represent the intensity/magnitude of precipitation in a given grid cell, the accumulated precipitation in mm ($p_a$) over the total duration of a cluster is used. Here, precipitation accumulation represents the total amount of precipitation accumulated over the duration of a cluster over one grid cell, including timesteps when the precipitation value is inferior to the 99th percentile threshold. To retain a single value characterizing a cluster, the largest value of $p_a$ among all the grid cells

included in a cluster is retained.

     b. Peak wind gust (w). The peak wind gust is the maximum wind gust over a grid cell over the duration of a cluster. The intensity of a wind cluster is expressed by the maximum peak wind gust in the cluster duration in m s$^{-1}$.

(ii) *Spatiotemporal attributes*.

a. The spatial extent is measured in grid cells ($0.25° \times 0.25°$). It represents the total number of grid cells (**Fig. 2**) involved in the cluster.

     b. The temporal extent (or duration) is measured in hours. The temporal extent represents the difference between the last timestep and the first timestep in which the cluster occurs.

(iii) *Historical attributes.* These attributes include the following:

a. the start and end date of an event,

     b. Season: Dec./Jan./Feb. [DJF], Mar./Apr./May [MAM], June/Jul./Aug. [JJA], Sep./Oct./Nov. [SON]

     c. Location: grid cells involved in the cluster.

These attributes are summarised in **Table 1**.



**Table 1: Intensity and spatiotemporal attributes of hazard clusters and their availability for wind, precipitation and compound hazard events in the present study**

|  | Attribute | Wind clusters | Precipitation clusters | Compound wind–precipitation clusters |
|---|---|:---:|:---:|:---:|
| **Intensity** | $p_a$ (mm) |  | ✔ | ✔ |
|  | $w$ (m s$^{-1}$) | ✔ |  | ✔ |
| **Scales** | Spatial footprint (%) | ✔ | ✔ | ✔ |
|  | Duration (h) | ✔ | ✔ | ✔ |
| **Historical** | Start time (h) | ✔ | ✔ | ✔ |
|  | End time (h) | ✔ | ✔ | ✔ |
|  | Location (cells involved) | ✔ | ✔ | ✔ |

## 5 Results

This section presents the results of applying our SI–CH methodology to ERA5 precipitation and wind variables for 1979 to 2019 in the UK. From these attributes (**Table 1**), the distribution of scales attributes are presented and discussed along with other characteristics of the wind, precipitation and compound hazard database created (**Sect. 5.1**). Historical attributes of the hazard clusters created are confronted with a catalogue of 157 observed significant Great Britain weather events. This confrontation highlights not only our methodology's capabilities, but also the ability of the ERA5 reanalysis to detect different types of extreme events in Great Britain (**Sect. 5.2**). The scales and intensity attributes of detected clusters are then analysed with examples from the significant events catalogue (**Sect. 5.3**).

### 5.1 Wind, precipitation, and compound clusters identified using the SI–CH methodology

We apply the SI–CH methodology (**Sect. 4**) over the spatiotemporal dataset presented in **Sect**. 3 for January 1979 to September 2019, and detect a total of 18,086 precipitation clusters, 6190 wind clusters, and 4555 compound hazard clusters. The detailed attributes for these single and compound hazard clusters are given in the ERA5 Hazard Cluster Databases (**Supplement 3**), including all attributes given in **Table 1**.

Ten examples of clusters of various sizes and durations detected by the SI–CH methodology are displayed in **Fig. 8**. For each type of cluster (precipitation, wind, compound) the footprint of one small, one medium and one large cluster are presented. These examples illustrate the diversity of shape, area and duration of wind and precipitation clusters detected. For compound hazard clusters (**Fig. 8g, 8h, 8i**), different configurations are displayed: a small compound hazard cluster at the intersection of two large precipitation and wind clusters (**Fig. 8g**), a small precipitation cluster contained within a large wind cluster (**Fig. 8h**) and a large wind cluster associated with two precipitation clusters (**Fig. 8**), creating two distinct compound hazard clusters.





**Figure 8: Footprints of ten example natural hazard clusters from ERA5 Hazard Clusters Database (Supplement 3): three precipitation clusters (a, b, c), three wind clusters (d, e, f), and four compound hazard clusters (g, h, i) detected by the SI–CH methodology proposed in this paper. The cluster ID (P = precipitation, W = wind, C = compound) is given at the top of each graph. The compound clusters shown include one P and one W cluster: (g) C233 = P1162 & W387; (h) C141 = PR766 & W220; (i) C2600 = P10,041 & W3717; (i) C2601 = P10,042 & W3717. W3717 is shared by both compound clusters C2600 and C2601. Clusters with areas that are small (footprint < 9 cells) are shown in the left column (a, d, g), medium (19 cells<footprint<32 cells) in the middle column (b, e, h), and large (footprint>316 cells) in the right column (c, f, i). The definition of small, medium and large for single and compound hazard clusters is derived from the quantiles of the footprints' distribution (q10, q50, q95). Circle size represents the duration of single or compound hazard clusters in each cell.**





395  In our SI–CH methodology, we decided to have each compound cluster comprised of just two clusters: one precipitation and
one wind cluster. Therefore, an extreme precipitation (or wind) cluster with the same ID can be part of two (or more) different
compound hazard clusters as displayed in **Fig. 8i**. These compound clusters might overlap in time and/or space. The 4555
compound hazard clusters we detected are composed of 3565 precipitation clusters with a unique ID (20% of the 18,086 single
hazard precipitation clusters) and 2913 wind clusters with a unique ID (47% of the 6190 single hazard wind clusters). For

400  example, an extratropical cyclone bringing extreme precipitation scattered in space and time could be identified as several
compound hazard clusters composed of different precipitation clusters and one single extreme wind cluster. In our database of
4555 compound hazards clusters, we found the following distribution of unique single event hazard cluster IDs:

- Of the 3565 precipitation clusters with a unique ID, 2912 (82%) are each found in 1 unique compound cluster, 578
  (16%) in 2–3 different compound clusters, and 75 (2%) in 4–9 different compound clusters.

- Of the 2913 wind clusters with a unique ID, 2053 (70%) are found in 1 unique compound cluster, 663 (23%) in 2–3
  different compound clusters, 156 (5%) in 4–5 different compound clusters, and 41 (1%) in 6–14 different compound
  clusters

Regarding the distribution of exclusive vs non-exclusive single hazard clusters making up the compound clusters, where
exclusive means a unique single-hazard ID, wind or precipitation, is found in only one compound hazard ID, we found the

following:

- **Non-exclusive wind ID** and **non-exclusive precipitation ID** clusters: 559 (**12%**) of compound clusters.
- **Non-exclusive wind ID** and **exclusive precipitation ID** clusters: 1943 (**43%**) of compound clusters.
- **Exclusive wind ID** and **non-exclusive precipitation ID** clusters: 1084 (**24%**) of compound clusters.
- **Exclusive wind ID** and **exclusive precipitation ID** clusters: 969 (**21%**) of compound clusters.

**Figure 9** presents the probability distributions of duration (h) and spatial footprint (% of the study area) for the 4555 compound,
18,086 precipitation and 6190 wind clusters. The diamond for each violin plot represents the average of the values for the
given variable. Precipitation, wind and compound clusters vary in shape, size and duration. In **Fig. 9a**, we observe that for the
spatial footprint, wind clusters (9.0%) are on average larger than precipitation (5.0 %) and compound clusters (4.6 %). Spatial
footprints range from one grid cell, representing <0.1% of the study area, to 100% of the study area for wind and precipitation

clusters and 89% for compound hazard clusters. The duration (**Fig. 9b**) of single and compound hazard clusters varies from 1
h to 4 days, with compound clusters lasting on average 24 h, which is much greater than wind (average 9.6 h) and precipitation
(average 6.8 h) clusters). Only 2.4% of precipitation clusters have a duration greater than 24 h compared to 8.8% of wind
clusters and 43.5% of compound hazard clusters. The long duration of compound hazard clusters can be explained by the
definition of compound hazard events presented in **Sect. 4.3**. **Figure 9** highlights the capacities of our approach to adapt to

different input data.

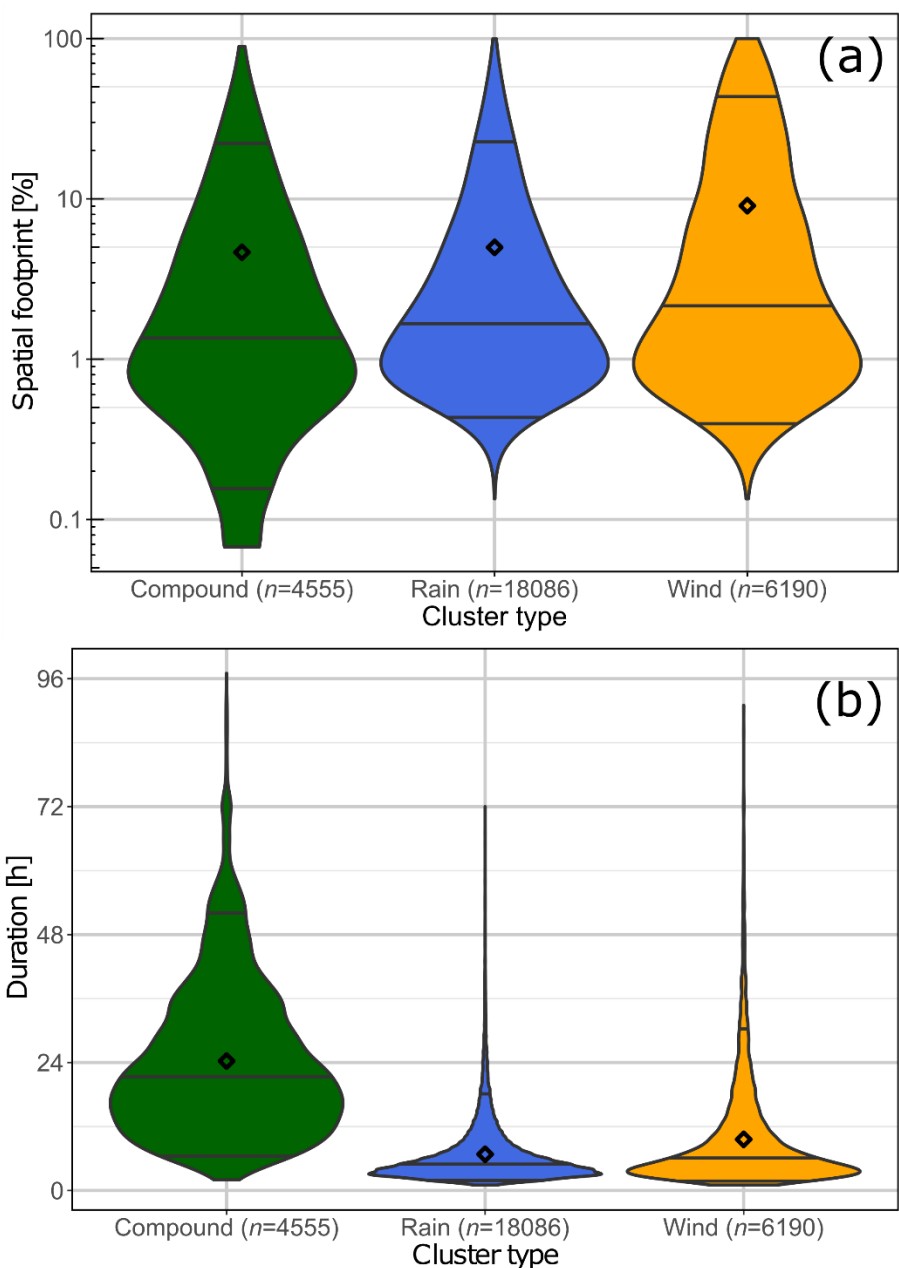

**Figure 9: Violin plots for 4555 compound, 18,086 precipitation, and 6190 wind clusters, for January 1979 to September 2019 from the ERA5 Hazard Cluster Database (Supplement 3). Shown are (a) spatial scale as a percentage of the total study area and (b) duration in hours (h). Black diamonds represent the mean of the distributions. Quantiles 0.05, 0.5 and 0.95 are also displayed with horizontal lines. See Fig. 8 for ten examples of clusters.**




## 5.2 Event identification: Confrontation with significant events

To assess the capacity of our methodology to identify observed hazard events, natural hazard clusters from the ERA5 Hazard Clusters Database (**Supplement 3**) are confronted with a set of past significant hazard events that impacted Great Britain. To do so, we created a catalogue of 157 significant Great Britain weather events that occurred between January 1979 and
September 2019 (see **Supplement 4**). These 157 major events selected aim to represent the broad range of events, including extreme precipitation and/or extreme wind impacting Great Britain. The construction of the catalogue is done using four primary sources:

- *British Weather Disasters (1901–2008)* (Eden, 2008). Chronology of severe weather events in the UK.
- *Global Active Archive of Large Flood Events (1985–present)* (Brakenridge, 2021): An archive of flood events derived
from news, governmental, instrumental, and remote sensing sources.
- *EM-DAT (Emergency Events Database) (1984–2020)* (CRED, 2020): A record of disasters maintained by the Centre for Research on the Epidemiology of Disasters (CRED).
- *Past weather events website (1990–2020)* (Met Office, 2020) : Archive of reports on past weather events from the UK Met Office.

These sources do not focus exclusively on extreme precipitation and wind events. Therefore, creating our significant Great Britain weather events catalogue involves a pre-selection based on the event's relevance to the study. For inclusion, we used the following criteria for inclusion in our catalogue:

- An event must include extreme precipitation and/or extreme wind (the source mentions that it is extreme, which is often relative to the source/location).
- The event duration must not exceed 5 days (which is above the maximum duration of clusters detected by the SI–CH method, e.g., see **Fig. 8b**). For example, this removed events which were 'extreme precipitation/flood' events recorded as occurring over weeks or months, where the source did not separate precipitation duration and flooding duration. For these, the same event with the duration of the extreme precipitation event was found from another source, and included (as they were normally ≤5 days). Overall, <10% of 'extreme' events were removed for having
a time duration > 5 days.
- Where multiple sources identified the same event, the authors made a judgment on which source had the most accurate representation of that event.

Most of the extreme events, according to the sources, were selected, with an emphasis on events recorded in more than one source. Particular attention was given to the selection of events of various size and duration. The four sources are used to
identify each event's timing, location and duration. Duration is expressed in days, while each event's location corresponds to the eleven NUTS1 regions of Great Britain (ONS, 2021): Northeast (England), Northwest (England), Yorkshire and The Humber, East Midlands (England), West Midlands (England), East of England, London, Southeast (England), Southwest



(England), Wales and Scotland. An event can occur over one or more NUTS1 regions. Significant events are also characterised by their dominant hazards (the primary hazard reported in the sources).

Events per year for 1979–2019 are divided into precipitation events ($P$) and wind events ($W$), depending on their dominant hazard as given in the four databases above (Eden, 2008; CRED, 2020; Met Office, 2020; Brakenridge, 2021). Some significant events also include associated hazards (e.g., landslides) when reported by the sources. We use these sources to compile a significant Great Britain weather events catalogue for 1979 to 2019 which contains 96 extreme precipitation events ($P$) and 61 extreme wind events ($W$) and is given in its entirety in **Supplement 4**. **Figure 10** shows the date and region of occurrences of

the 157 significant events in our catalogue: 96 extreme precipitation events (heavyweight blue circles) and 61 extreme wind events (heavyweight orange crosses). Of the 157 events in the significant weather events catalogue, 24 can be considered as compound hazard events (lightweight green circle overlain by a cross) where the extreme wind and extreme precipitation are both reported in **Supplement 4**. As mentioned previously, events can occur in one or more NUTS1 region. Of the 157 catalogue events, 63 (40%) are in one NUTS1 region, 29 (18%) in two NUTS1 regions, 23 (15%) in three NUTS1 regions, 18 (11%) in

four to six NUTS1 regions, and 24 (15%) in ten to eleven NUTS1 regions. These latter are events covering the majority of Great Britain.





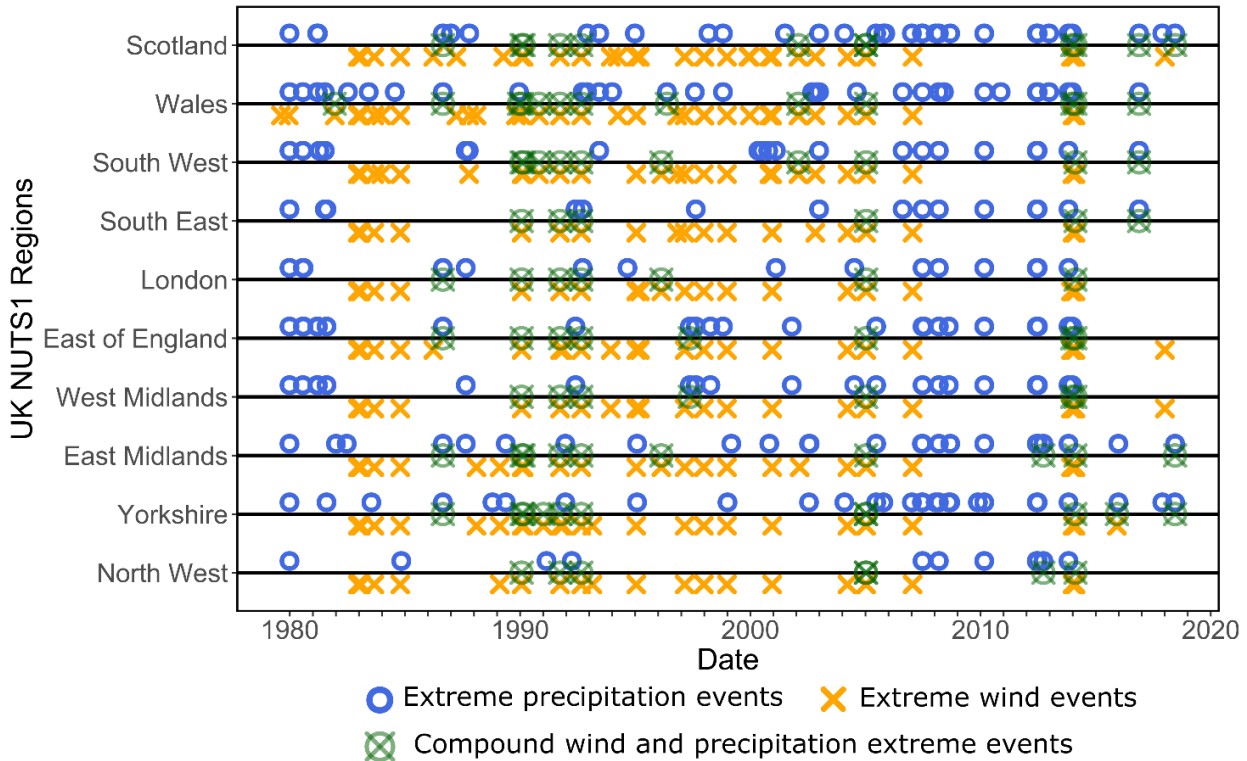

**Figure 10: Timeline of 157 events in our significant Great Britain weather events catalogue (Supplement 4, 1979 to 2019) used to assess the detection abilities of the SI–CH methodology of the 11 NUTS1 regions of Great Britain. Significant events are considered as *precipitation* (96 events, heavyweight blue circles) or *wind* (61 events, heavyweight orange crosses). Of the 157 events in the significant weather events catalogue, 24 can be considered compound hazard events (lightweight green circle overlain by a cross) as given in the sources.**

In **Fig. 10**, we observe the interconnections between regions impacted by the same events (e.g., January 2010 precipitation event) and the clustering of events in time. Some regions are also more represented than others in our catalogue. The number of events per regions is displayed in **Fig. 11a**, with South-West England and Wales being the regions with the most events and North-East and East England being the regions with the fewest events.

The date and locations of the 157 events are then used to assess our clustering method's ability to capture extreme wind or extreme precipitation events. For each event in the significant Great Britain weather events catalogue (**Supplement 4**), a temporal and spatial match is performed to identify the corresponding cluster(s) in our ERA5 based variable results from **Sect. 5.1**. There are eleven NUTS1 regions. For a spatiotemporal match to occur, the following need to be true:

- A cluster needs to occur in the same NUTS1 region(s) as the significant weather event catalogue event.
- A cluster needs to occur during the same day(s) as a significant weather event from the catalogue (**Supplement 4**).

The hit rate (ratio between the number of events with corresponding clusters and the total number of events) is used to assess the capacity of the SI–CH methodology. Over Great Britain, 147 out of 157 (hit rate = 93.4%) significant events have one or more corresponding hazard clusters, when spatial and temporal matching is done. The hit rate is slightly higher for the subgroup



of extreme wind events (95.1%) than for extreme precipitation events (92.6%). Among these 147 events, 64 (43.5%) have one corresponding cluster. The percentage of detected events for each NUTS1 region varies between 91.7% (South-East England) and 100% (North-West England, North-East England) and is displayed in **Fig. 10b**.Among the 147 events with clusters

associated, 109 are identified as compound hazard events (21/24 for compound hazard events reported in the catalogue). This suggests that compound hazard events are underreported on our catalogue (**Supplement 4**).

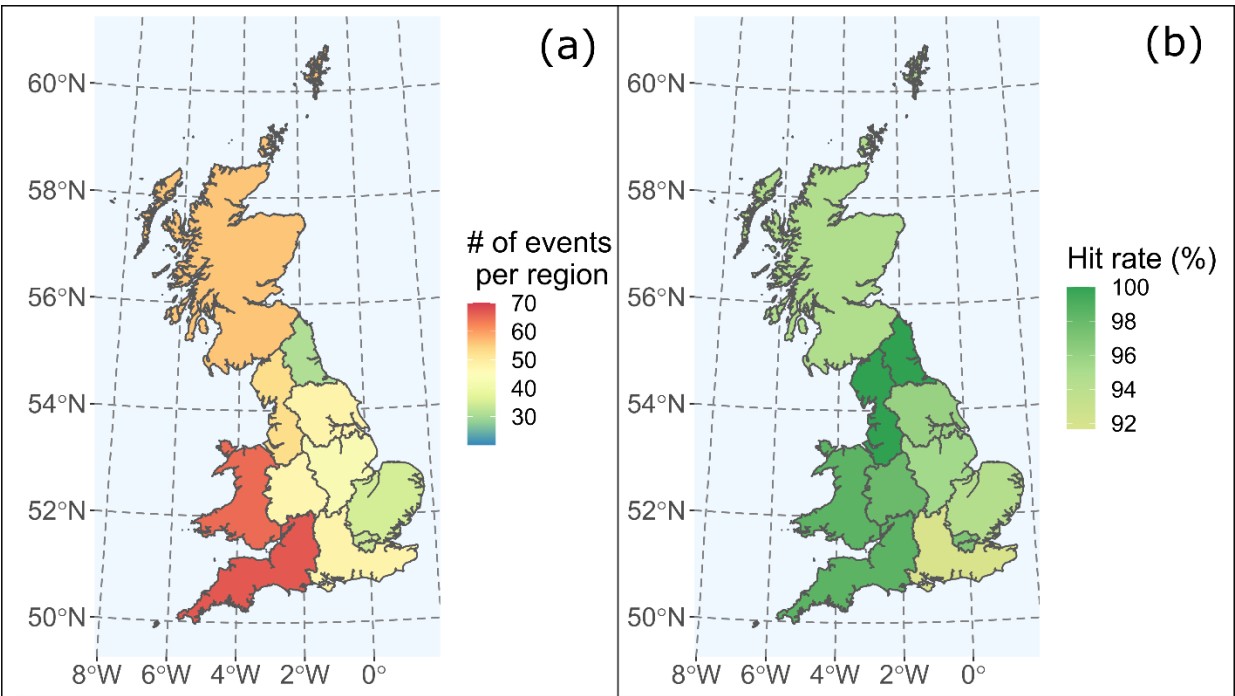

**Figure 11: Map of Great Britain divided into 11 NUTS1 regions showing: (a) the number of events per region from our significant Great Britain weather events catalogue (Supplement 4) and (b) the hit rate (ratio between the number**

**of joint events and the total number of events in our significant weather events catalogue for each region**

**5.3 Spatiotemporal properties of compound wind and precipitation extremes in Great Britain**

Only a minority of the single and compound hazards clusters detected during 1979–2019 can be associated with events that led to considerable damages (e.g., Great Storm of 1987, Storm Xaver). We now illustrate the SI–CH methodology using three examples from the significant events catalogue. Spatiotemporal properties of single and compound hazard clusters in relation

to hazard intensity are then discussed.

The intensity of precipitation and wind events is assessed with the intensity attributes presented in **Table 1.** Values of precipitation accumulation and peak wind gust are subject to uncertainties (**Sect. 3**). Therefore, precipitation accumulation and wind gust duration values are transformed onto the standard uniform space on the interval [0,1]. The empirical cumulative probability expresses the intensity of hazards for both wind accumulation and precipitation duration as the following:





$$P(x_i) = \frac{R_{x,i}}{N_x + 1} \tag{3}$$

where $R_{x,i}$ represents the rank of observation $x_i$ in the sorted sample ($i = 1$ for the smallest observation, $i = N$ for the largest), and $N_x$ is the sample size. For compound hazard clusters, the combined intensity is expressed by the minimum cumulative probability of the two hazards:

$$P(x_i, y_i) = \min \left( \frac{R_{x,i}}{N_x + 1}, \frac{R_{y,i}}{N_y + 1} \right) \tag{4}$$

where $R_{x,i}$ ($R_{y,i}$) represents the rank of observation $x_i$ ($y_i$) in the sorted sample ($i = 1$ for the smallest observation, $i = N$ for the largest), and $N_x$ ($N_y$) is the sample size.


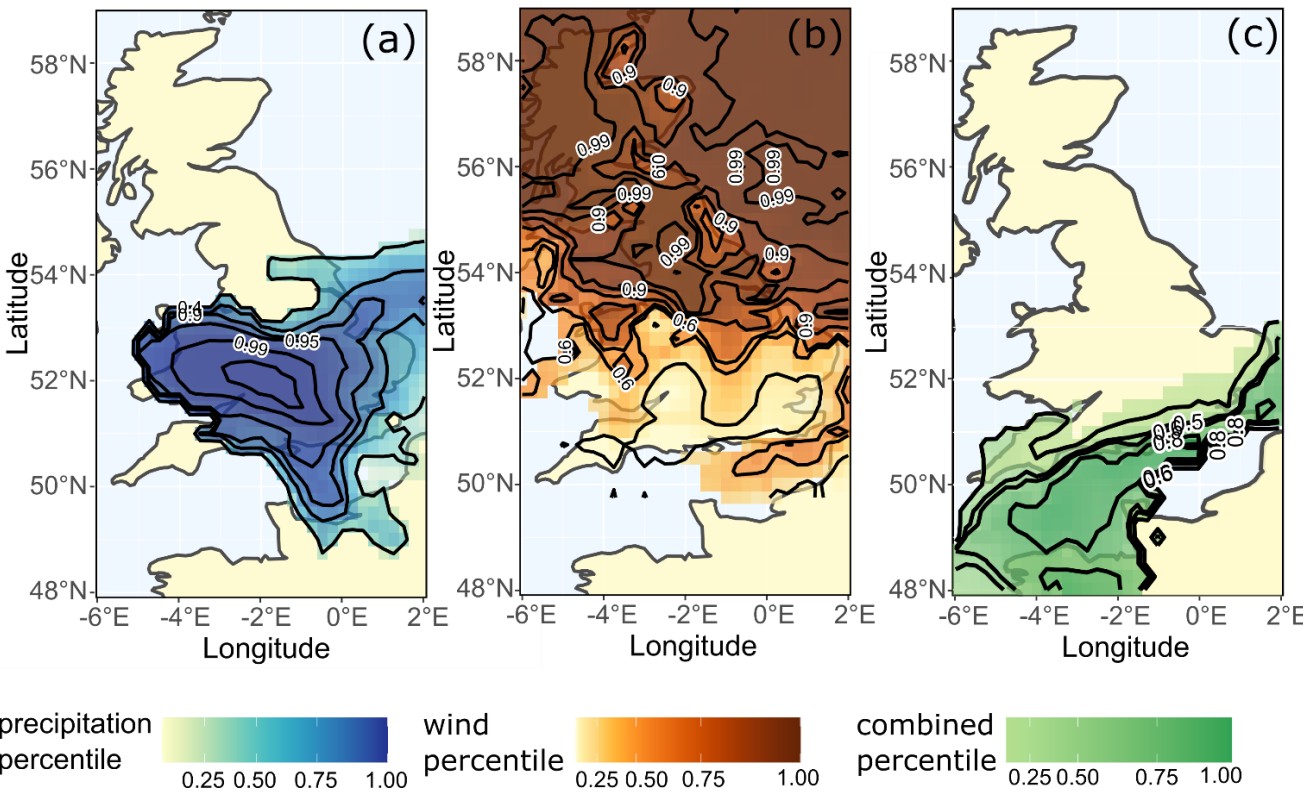

**Figure 12: Spatial footprint and intensity of precipitation and wind clusters (from the ERA5 Hazard Cluster Database, Supplement 3) associated with three significant events from our Significant Great Britain Weather Events Catalogue (Supplement 4). The spatial footprint as a function of intensity for all pixels for each of these three events is given: (a) Catalogue Event 130, extreme precipitation**
**(Hazard Cluster Database P12,593); (b) Catalogue Event 146, extreme wind, associated with Storm Xaver (Hazard Cluster Database W5423); (c) Catalogue Event 154, compound wind and precipitation event associated with Storm Angus (Hazard Cluster Database C4172).**





**Figure 12** highlights the spatial footprint of three hazard clusters and the intensity field of single and compound hazards within the clusters. **Figure 12a** shows the footprint of a precipitation cluster that occurred in July 2007 (**Supplement 3**, ERA5 Hazard

Cluster Database, P12593). The total footprint occupies the vast majority of Wales and southern England. However, the most intense precipitation values are confined to the West Midlands, where most flooding and impacts were reported (Eden, 2008). **Figure 12b** shows the footprint associated with Storm Xaver (December 2013), which develops over 73% of the study area with varying intensity (**Supplement 3**, ERA5 Hazard Cluster Database, W5423). The most extreme winds occurred in North-East England, Scotland and the North Sea. **Figure 12c** highlights the compound hazard footprint of the cluster associate with

Storm Angus (November 2016), where extreme precipitation and wind combine at high intensities, mostly over the British channel (**Supplement 3**, ERA5 Hazard Cluster Database, C4172).



**Figure 13: Spatial-quantile plot for 4555 compound, 18,086 precipitation, and 6190 wind clusters, for January 1979 to September 2019 from the ERA5 Hazard Cluster Database (Supplement 3). The spatial footprint (in % of the total area) is given as a function of the intensity of single and compound hazard events. The intensity (from 0.00 to 1.00) is expressed by (a) the cumulative probability (Eq. 3) of the precipitation accumulation at each grid cell during all precipitation events, (b) the cumulative probability (Eq. 3) of the wind 'accumulation' at each grid cell during all wind events and (c) the minimum cumulative probability (Eq. 4) of the compound hazard (wind + precipitation) events. Coloured curves represent clusters associated with events from the our Significant Great Britain Weather Events Catalogue (Supplement 4); colours represent the duration of the events. Grey lines represent all other events in each database (Supplement 3). The blue curve in each part represents the individual hazard event clusters for precipitation, wind, compound, displayed in Fig. 7.**



**Figure 13** displays the spatial footprint as a percentage of the study area displayed in **Fig. 3** of clusters as a function of their intensity. An intensity of $I = 0.00$ represents the minimum intensity value in our sample of extremes (1.42 mm for precipitation over the duration of the event over all event cells and 17.11 m s$^{-1}$ for wind over a given cell for an event). Clusters related to

events in the catalogue are highlighted in colours, while other events are grey. The three clusters displayed in **Fig. 12** are in dark blue **Fig. 13** for (a) precipitation, (b) wind and (c) compound wind and precipitation extreme clusters. It is important to note that compound hazard clusters in **Fig. 13c** are constructed with single hazard clusters from (a) and (b). In **Fig. 13**, each curve corresponds to a cluster and shows the evolution of the footprint (number of cells) as a function of their intensity. For example, cluster P12593 has a total footprint of approximately 28% of the study area with an intensity (I) above 1.42mm, this

footprint drops to 25% for an intensity above q50 (9.99 mm) and 4% for an intensity above q99 (40.14 mm). The colour of each curve represents the total duration of each cluster.

From **Fig. 13**, we also observe that the largest (highest spatial footprint at intensity=0.0) and most intense (highest spatial footprint at intensity=1.0) clusters are mostly associated with the 157 significant events presented in **Sect. 5.2**. This suggests that events from the catalogue developed in **Sect. 5.2** correspond with the most noteworthy clusters obtained from the SI–CH

methodology. However, there are several clusters with short-duration, small spatial footprint and moderate intensity associated with the 157 significant events, particularly for precipitation clusters (**Fig. 13a**). This is because more than one cluster can be associated with a significant event. For our cluster database, there are on average 2.8 clusters detected per significant event for extreme precipitation events, 1.6 clusters detected per significant events for wind events and 2.1 clusters per significant event for compound hazard events in the catalogue. In practice, a significant event can be associated with one large and/or intense

cluster and several small clusters. For example, this is the case for event 136, which is associated with large and intense precipitation cluster (P13,158) and two small and low-intensity precipitation events (P13161 and P13165).

## 6 Discussion

Assessing the characteristics of compound hazards events in space and time brings valuable insight into the nature of the relationship between the hazards involved in the event. It overcomes the main limitations of compound hazards studies which

focus on interrelations at specific sites (Sadegh et al., 2018). However, spatiotemporal analysis of compound hazards brings its own set of uncertainties and limitations. This section will discuss the following four main limitations arising from the presented study:

- parameters influencing the clustering procedure,
- the subjective definition of compound hazards events in space and time,

- uncertainties around the estimation of attributes and input data

*Parameters influencing the clustering procedure.* Three main parameters are influencing the clustering process and consequent results; their influence is discussed further and quantified in **Supplement 2**.



    (i)   The threshold ($u$) selected to sample extreme events. This study is based on the assumption that an extreme enough occurrence of an environmental variable can be used as a proxy for natural hazard identification. A threshold is then set to sample the extreme occurrences of environmental variables. Even if this threshold has been selected in light of previous works on wind and precipitation extremes (Ulbrich et al., 2009; Martius et al., 2016), its value remains subjective. A seasonal threshold could also have been used to detect more events during the extended summer. The value of the threshold directly impacts the number of extreme events sampled and, therefore, on the selection of the other clustering parameters (**Supplement 2**).

    (ii)  The ratio $r$ of the spatiotemporal scaling parameters $a$ and $b$. A three-dimensional Euclidean distance is used as a distance measure for the clustering procedure. The value of the distance between each extreme event is controlled by the importance given to the spatial (longitude and latitude) and temporal (time) component in the input data. Each component was set to have the same importance in the distance computation, but more importance could be given to the time (or space) component depending on a prior assumption (Zscheischler et al., 2013; Vogel et al., 2020).

    (iii) The density threshold $\mu$. While the neighbouring parameter $\varepsilon$ is set systematically (**Sect. 4.2**), its value depends on the density threshold, giving the minimum number of detected events per cluster. The selection of $\mu$ is based on a prior assumption about the minimum size a compound hazard event can have in the context of the study.

*The subjective definition of compound hazards events in space and time.* **Sect. 4.3** presented four different possible definitions for a compound hazards event in time and space. It was chosen to define the duration as the aggregated duration of all hazard events; however, one could be more interested in extracting the simultaneous duration of both hazards.

*Biases and uncertainties around the estimation of attributes.* There are biases and uncertainties around the values of intensity attributes of the events. These biases are partly due to the data used in this study: the ERA5 reanalysis data (**Sect. 2**). Higher uncertainty arise from the estimation of precipitation accumulation as precipitation observations are not assimilated in ERA5. Biases might also be more pronounced over mountainous areas (Skok et al., 2016; Sharifi et al., 2019) which are more exposed to compound wind and precipitation events (**Appendix A**). The size of the study area also leads to some events being detected only partially which could bias our estimates of size and duration of events.

## 7 Conclusion

To characterise more accurately the compound hazard event of extreme precipitation and extreme wind events in Great Britain, their overlap in space and time has been analysed. By clustering extreme occurrences of maximum hourly wind gust and hourly precipitation from ERA5, 4555 compound wind–precipitation clusters over Great Britain were identified for 1979–2019 (**Supplement 3**). To assess the ability of the approach to identify the occurrence of extreme events in time and space, a catalogue of 157 extreme precipitation and/or extreme wind events that occurred in Great Britain over the period 1979–2019 was created (**Supplement 4**). The confrontation was done at a regional (eleven NUTS1 regions) and daily scale. The average



hit rate (the ratio between the number of identified events and the total number of events) over the whole area is 93.7%,
      meaning that our approach successfully identifies most of the extreme precipitation and wind events. A total of 24 (15%) of
      the 157 events of the catalogue were reported as compound events (wind–precipitation). With the SI–CH methodology, we
      identified 109 compound hazard clusters to be associated with the 157 significant weather events (69%). The approach's
      potential to analyse the footprint and intensity of events was then highlighted by examining three events from the catalogue.

Additionally, the importance of the intensity of natural hazards within clusters is addressed, showing that some events develop
      over large areas with localised spots of extreme intensity. In contrast, other events have smaller but steady spatial footprints
      when increasing the intensity (e.g., precipitation cluster associated to event 130). The strength (ability to identify significant
      extreme events) and weaknesses (more than one cluster per significant events) are finally highlighted and discussed.

      One important limitation of our approach is its reliance on the input data. To estimate with more accuracy intensity attributes

(particularly for precipitation), one would require to use a statistical correction of the simulated precipitation (Widmann and
      Bretherton, 2000) or other gridded datasets based on observations (e.g., E–OBS). Reanalysis data have the potential to study
      compound hazard events as they offer homogenised values for an important number of variables. Our SI–CH approach coupled
      with ERA5 data has shown its ability to identify significant single and compound hazard events and allows the analysis of the
      spatial and temporal attributes of such events. The sequencing of hazard events can also be analysed with this SI–CH approach.

For example, the ERA5 Hazard Cluster Database (**Supplement 3**) created in this study could be used to identify sequences of
      single and compound hazard events (e.g., extratropical cyclones sequences).

      Finally, the SI–CH approach can be extended to the analysis of other compound events such as compound hot and dry events
      (Sutanto et al., 2020) and compound cold and snow events (Hillier et al., 2020). The definition of the compound hazard in time
      and space as proposed in this paper can also be extended to more than two hazards. This allows the methodology to be

potentially extended to the identification of more complex compound events, such as compound hot-dry event with extreme
      wind and extreme heat, drought and wildfires.





### Appendices

**Appendix A: Complementary analysis of spatiotemporal features of compound wind–precipitation clusters**

**Summary:**

This appendix consists of a complimentary analysis (with figures) on spatiotemporal features of compound wind–precipitation clusters in Great Britain and highlights how the database of compound hazard clusters can be further exploited. In this appendix, we will present six figures:

- The *proportion of compound hazard clusters* among *wind and precipitation clusters* is analysed with respect to:
  - The size and duration of these clusters (**Fig. A1**).
  - Their location (**Fig. A2**).
- The *frequency of occurrence* of compound wind–precipitation events over Great Britain is estimated, allowing the identification of compound wind–precipitation hotspots (**Fig. A3**).

- The *strength of the spatiotemporal dependence between precipitation clusters and wind clusters* is assessed through the Likelihood Multiplication Factor (LMF) (**Fig. A3**).
- The *seasonality* of wind, precipitation and compound hazard clusters is analysed (**Fig. A4**).
- The *monthly frequency of compound hazard clusters* amongst the total number of clusters is analysed (**Fig. A5**).
- The *spatial dependence* between different sites is investigated (**Fig. A6**).


**Figure A1** shows the proportion of compound clusters amongst wind and precipitation clusters conditioned on the footprint (a) and duration (b) of clusters. The proportion of precipitation clusters and wind clusters involved in a compound cluster increases with the footprint when the cluster footprint is above 1% of the study area. For example, when considering all footprint sizes, compound hazards clusters represent 20% of all precipitation clusters (**Fig. A1**). For clusters with a footprint

greater than 10% of the study area (i.e., regional and multi-regional), the share of compound cluster surges to 52%.





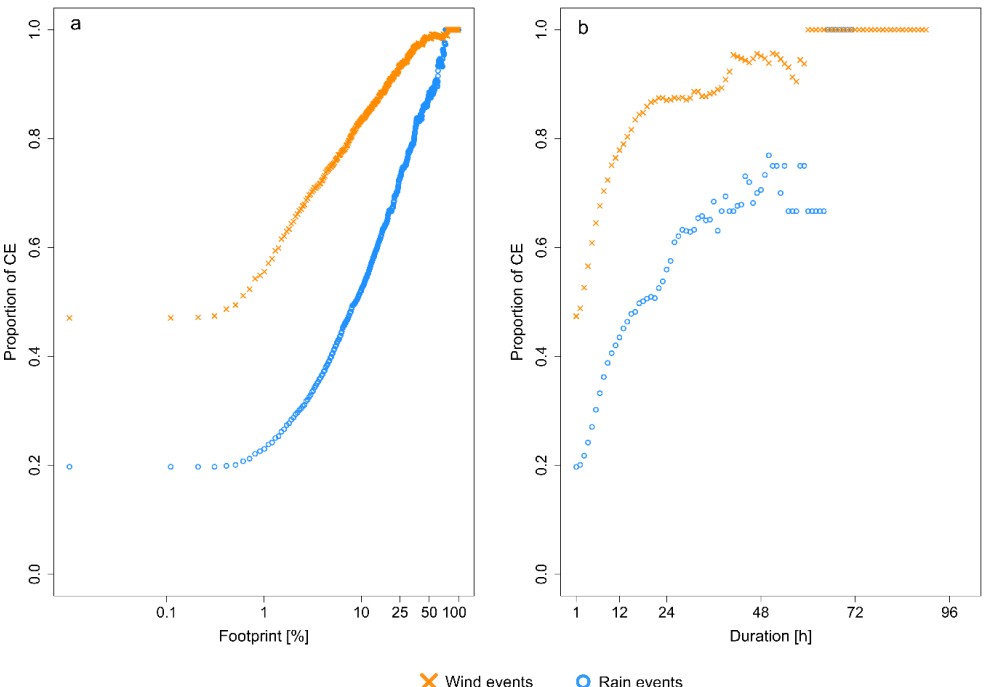

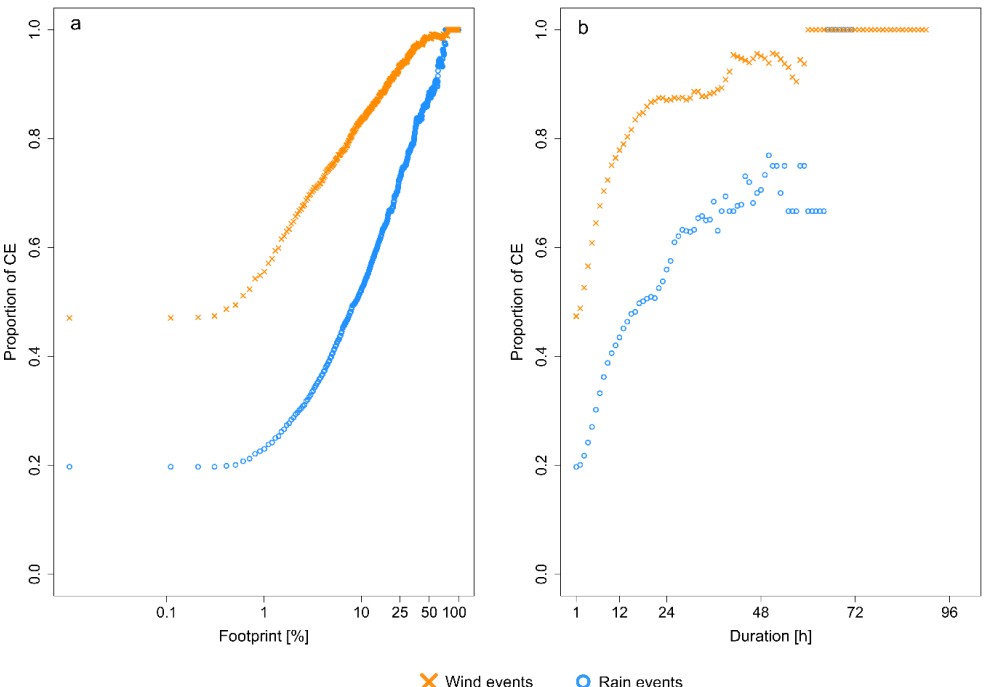

**Figure A1: Proportion of compound wind–precipitation clusters among wind clusters (orange) and precipitation clusters (blue) depending on (a) spatial footprint (b) duration of the hazard clusters.**

Over the study area, the proportion of compound wind–precipitation clusters among the precipitation clusters detected is 20%,

while 47% of the wind clusters are compound hazards clusters. However, this proportion is variable across Great Britain. **Figure A2** displays the fraction of compound hazard clusters among (a) wind clusters and (b) precipitation clusters. It highlights the spatial variability of compound cluster prevalence. Among the geographical features that may influence the frequency of compound hazards clusters among precipitation and wind clusters, orography probably plays an important role. The frequency of compound wind–precipitation clusters is the highest in mountainous areas while lowlands of the west coast

have a much lower frequency of compound wind–precipitation clusters among both precipitation and wind clusters.





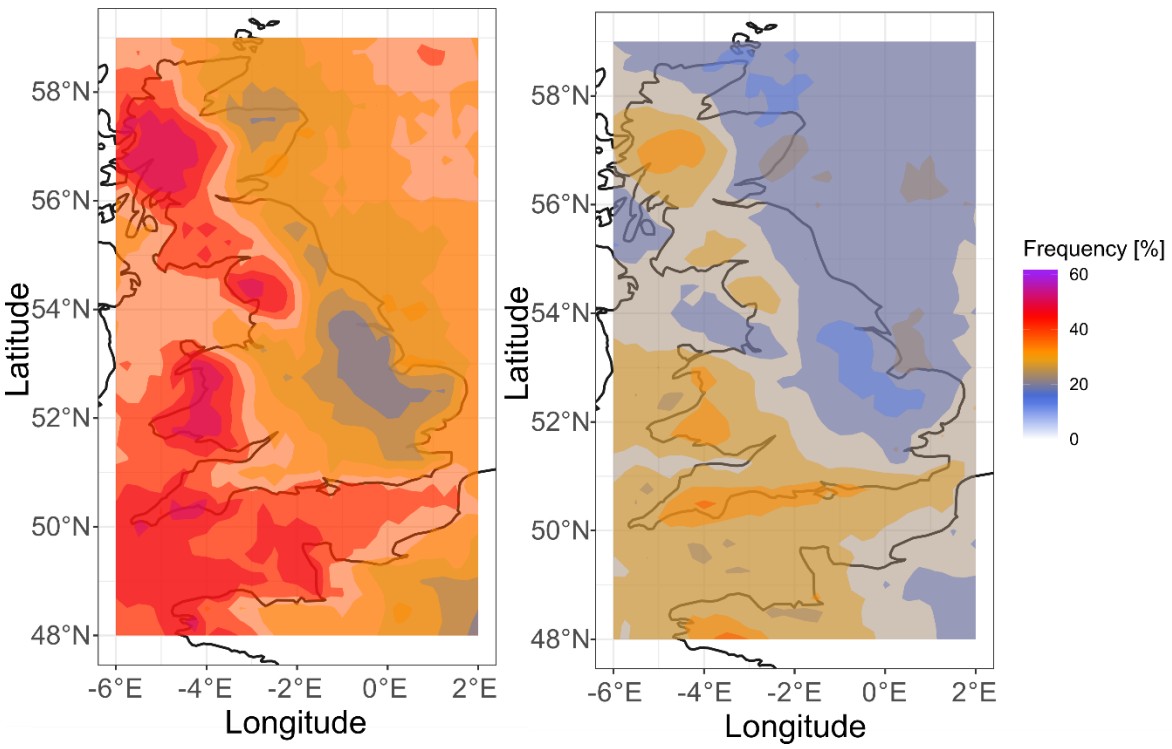

**Figure A2: Compound hazard (wind–precipitation) clusters proportion among (a) wind clusters and (b) precipitation clusters during the period 1979–2019 in Great Britain. Data from ERA5 (Hersbach et al., 2020).**

However, compound wind–precipitation clusters are more prevalent among the most intense hazard clusters. The latter

represents 58 of the 100 most intense precipitation clusters and 95 of the 100 most intense wind clusters. The intensity of precipitation and wind clusters is assessed with the intensity attributes presented in **Table 1**. The proportion of compound wind–precipitation clusters also increase with duration and footprint for both precipitation and wind clusters (**Fig. A1**).

As the duration of compound wind–precipitation clusters highly varies, their frequency of occurrence in the study area is

assessed by counting the number of hours in a compound cluster (as defined in **Section 4.3**) at each grid cell. The average number of hours per year in a compound cluster for 1979–2019 is displayed in **Fig. A3a**. This value varies between 20 and 95 hours in the study area. **Figure A3a** highlights regions that are more likely to be affected by compound wind–precipitation clusters with hotspots in mountainous area (as for **Fig. A2**). Nevertheless, the south-east coast of Great Britain is the primary hotspot for compound wind–precipitation clusters. The frequency of compound clusters gradually decreases eastward from

Cornwall and Wales toward Anglia and East Midlands showing a west-east decreasing gradient across all Great Britain. A similar pattern has been found for extreme precipitation (Blenkinsop *et al.*, 2017) and compound flooding (Hendry *et al.*, 2019). The prevailing direction of cyclonic weather systems and orography partly explains this pattern for compound wind–precipitation clusters (Hulme and Barrow, 1997).



The dependence between extreme wind and extreme precipitation (*w, p*) can influence the estimation of the joint return period. The influence of the dependence between extreme wind and extreme rainfall cluster occurrence is quantified using the likelihood multiplication factor (LMF) (Zscheischler and Seneviratne 2017). The LMF is the ratio between the joint return period considering the two variables dependent ($T_{dep}$) and independent ($T_{ind}$) of each other (Manning *et al.*, 2019):

$$LMF = \frac{T_{ind}}{T_{dep}} \tag{A1}$$

The likelihood multiplication factor (LMF) quantifies the influence of the dependence between wind clusters and rain clusters on the estimation of the frequency of compound wind–precipitation clusters (**Fig. A3a**). The LMF (**Fig. A3b**) shows the strength of the dependence between wind and rain clusters. The LMF > 1.0 in all parts of the study area, suggesting that rain and wind clusters do not occur independently. The LMF is particularly high along the south coast of Great Britain, in the British Channel and North West France. While occurrences of compound wind–precipitation clusters exhibit an east–west

pattern, the strength of the dependence between wind and rain hazard clusters has a South–North pattern.

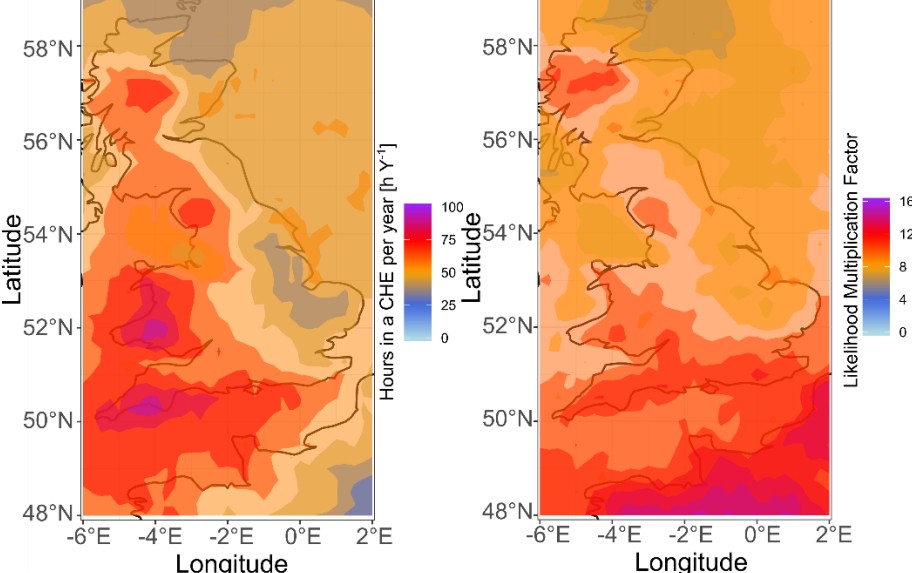

**Figure A3: Hotspots for compound wind–precipitation clusters in Great Britain. Showing (a) average number of hours in a compound hazards cluster in a year for 1979–2019 and (b) the likelihood multiplication factor (LMF) that quantifies the influence of the dependence between wind and rain cluster on the estimation of the probability of occurrence of compound hazards clusters.**
**Data from ERA5 (Hersbach et al., 2020).**

The spatial features of compound wind–rain clusters have been identified in **Fig. A2** and **Fig. A3**. Spatial disparities in their frequency and in the dependence between wind and rain clusters have been highlighted. These features also vary in time and with seasons. To look at the seasonality of single (wind only, rain only) and compound hazard clusters, all hazard clusters have been taken into account and divided into three categories: wind, rain and compound. Wind clusters that are part of a compound

cluster are removed from the category "wind" while rain clusters that are part of a compound cluster are removed from the





category "rain". Monthly occurrences of these three categories of clusters are displayed in **Fig. A4**. While occurrences of wind and compound clusters are correlated, with a high season in extended winter (ONDJFM) and a low season in the extended summer (AMJJAS), rain clusters occurrence follows an opposite pattern with a high season in AMJJAS and a low season in ONDJFM. Around 82% of all recorded compound hazard clusters occur during the extended winter.

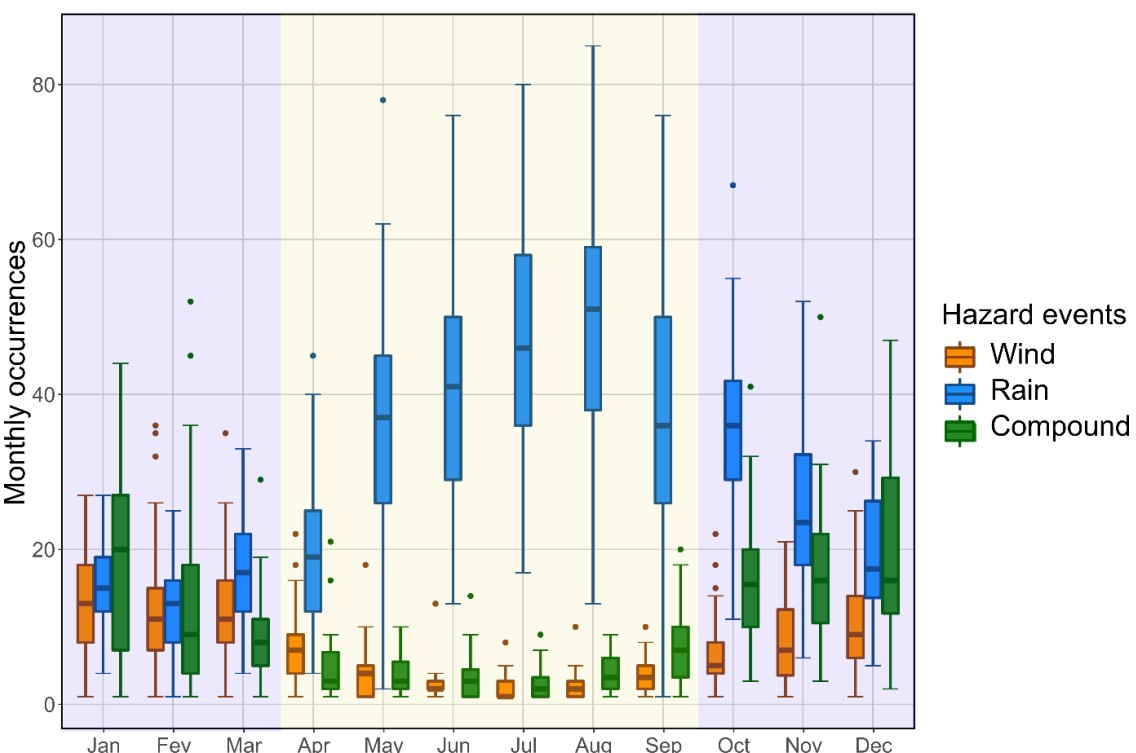


**Figure A4: Boxplots of the monthly number of wind (dark orange), rain (blue), and compound (green) hazard clusters in Great Britain over the period 1979–2019. Background colours represent the two seasons. Data from ERA5 (Hersbach et al., 2020).**

**Figure A4** provides a perspective on the seasonality of compound wind–precipitation clusters. It displays the proportion of compound wind–precipitation clusters among all clusters, with a seasonal proportion pattern similar to the one observed in

**Fig. A4**. This suggests that extreme rainfall and extreme wind clusters are more likely to co-occur during the extended winter. One possible explanation is that conditions leading to compound wind–precipitation clusters occur during the extended winter (Hillier *et al.*, 2020). This season coincides with the extra-tropical season in western Europe (Mailier *et al.*, 2006; Ulbrich *et al.*, 2009; Deroche *et al.*, 2014). Extra-tropical cyclone can bring several hazards including strong wind, storm surge, heavy rainfall and high waves (Frame et al., 2017). The influence of cyclonic weather systems coming from the Atlantic on

precipitation and wind extremes in Great Britain has been highlighted in previous pieces of work (Hawcroft *et al.*, 2012; Dowdy and Catto, 2017). **Figures A4**, **A5** and **A2a** suggest that such systems also have an influence of compound wind and precipitation extremes. However, this does not mean that every compound hazard cluster occurring during the extended winter





is an extratropical cyclone but suggests that such weather systems are drivers of compound wind–precipitation extreme

clusters**.**


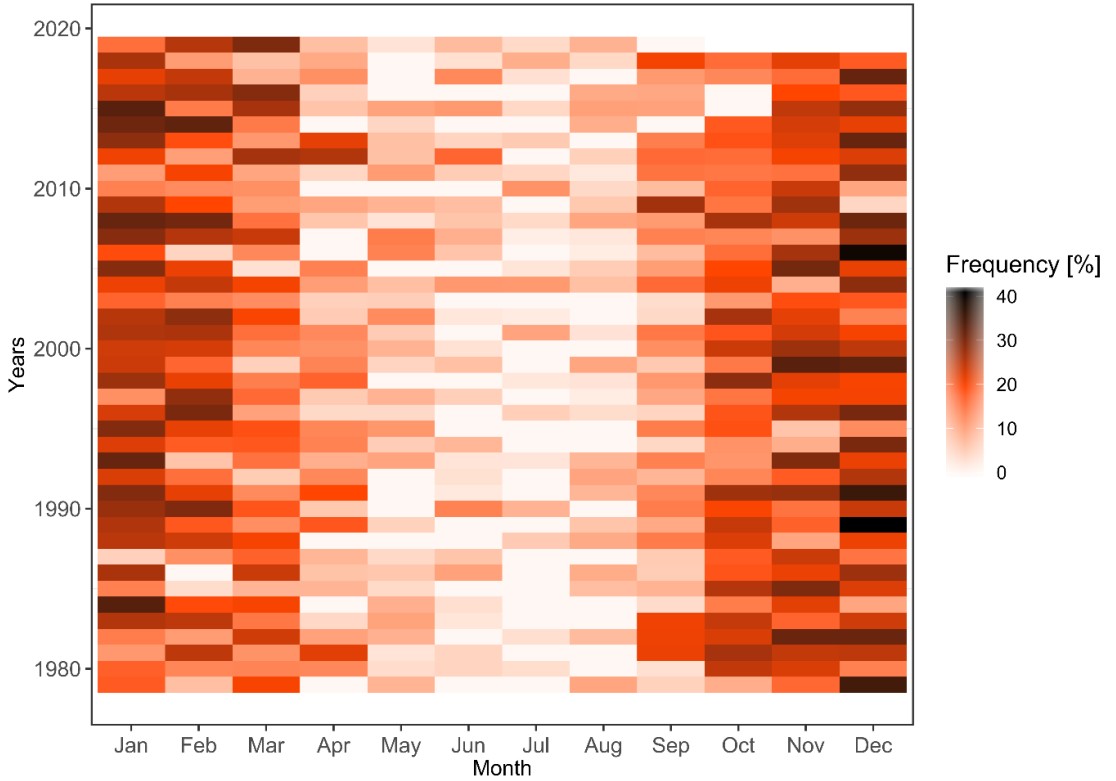

**Figure A5: Monthly fraction of compound hazards clusters among the total number of clusters (wind only+rain only+compound clusters) for that month for 1979–2019 over the study area. Each tile represents a pair for month-year; darker tiles meaning that the fraction of compound hazards clusters is greater.**


The spatial dependence between different sites is investigated in **Fig. A6.** This figure highlights each grid cell's probability in the study area to be in a compound wind–precipitation cluster, knowing that a given cell of reference is in a compound cluster (displayed as *Ps* in **Fig. A6**). Four locations in Great Britain are taken as cells of reference: Cumbria, Sheffield, London and Glasgow. The spatial extent of compound wind–precipitation clusters is displayed with a different perspective from the one

adopted in **Section 5**, highlighting that London is more likely to be in a large-scale clusters than Glasgow. Spatial dependences between places are also visible, for example, compound clusters occurrence in London is associated to compound clusters occurrence in South England while compound cluster occurring in Sheffield are more likely to develop over the Midlands and Wales.



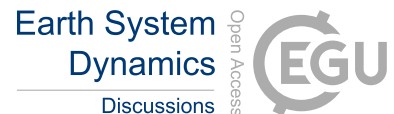

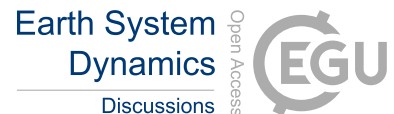

**Figure A6: Spatial dependence of compound wind–rain occurrence between different sites.** *Ps* **is The probability of each grid cell in the study area to be in a compound wind–precipitation cluster, knowing that a given cell of reference is in a compound cluster.**



**Code availability**

The codes used to generate the single and compound hazard clusters in this study are publicly available on GitHub: https://github.com/Alowis/SI-CH.

**Data availability**

We acknowledge the ERA5 datasets from ECMWF (https://www.ecmwf.int/en/forecasts/datasets/reanalysis-datasets/era5. Input, intermediary and output data used in this article are available at:

[https://zenodo.org/badge/DOI/10.5281/zenodo.4906264.svg].

**Supplement link**

There are 4 supplements to this article:

Supplement 1: Supplementary information on the DBSCAN algorithm

Supplement 2: Sensitivity Analysis of the spatiotemporal clustering procedure (SI–CH)

Supplement 3: ERA5 Hazard Cluster Databases (3.1. Wind, 3.2. Precipitation, 3.3. Compound hazards)

Supplement 4: Catalogue of significant Great Britain weather events catalogue

**Author contributions**

All authors discussed the whole plan of this article. AT and AJL designed the methodology. AT implemented all the analysis, prepared all the data, and finished the draft, including all figures in the article. BDM revised the article.

**Competing interests**

**Acknowledgements**

The first author was supported by an EDF R&D PhD studentship.

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
