# Peer review of "A Methodology for the Spatiotemporal Identification of Compound Hazards: Wind and Precipitation Extremes in Great Britain (1979–2019)"

_Earth System Dynamics, 2021_

## Referee Comment (RC1)

**Referee report:**

A Methodology for the Spatiotemporal Identification of Compound Hazards: Wind and Precipitation Extremes in Great Britain (1979–2019). Tilloy et al.

**1   Overview**

The authors present a method to identify compound hazards, in both space and time. The description of the method is very detailed and precise. A large catalogue of observed wind-precipitation hazards is additionally provided. It is used to assess the hazards identified by the algorithm applied on wind and precipitation data from the reanalysis dataset ERA-5.

The appendix also contains several results that would deserve to be interpreted more in detail, and summarized in the article itself. The hit rate between events from the catalogue and events identified by the algorithm is quite high. However, it would be nice to see a discussion on the events that were not captured.

This study is relevant and well described, this article deserves publication after minor revisions.

**2   Comments**

1. l. 8 "Compound hazards are two different natural hazards". Actually two or more, as written in conclusion l.629

2. Even if you are not using it, consider mentioning methods of threshold selection, in the context of spatio-temporal clustering (e.g. Kholodovsky and Liang, 2021)

3. l.50 space missing between period and 1979-2019.

4. l.56 the paragraph is a repetition of the previous one.

5. l.67 repetition e.g.

6. Maybe don't introduce acronyms/variables if you don't use them more than twice (CHE l.87, $x$ and $y$ l.90)

7. l.171 You mention the literature regarding the quality assessments of ERA-5 precipitation, but not on ERA-5 wind. Even if there is none, it can be worth mentioning it.

8. Sometimes the word "footprint" refers to spatial footprint only, sometimes spatio-temporal footprint. I would recommend being consistent, and maybe always write "spatial footprint" when it is only about spatial considerations.

9. It should be "Fig- 5."

10. A general remark about the figures: when plotting continuous variables, instead of qualitative color maps, I would highly recommend using sequential color maps, with homogeneous brightness gradient. Consider changing color maps in figures 5., 11a., A2, A3 and A6.

11. l.259 Please motivate a bit more the choice of $r = a/b$ (earlier or/and in more details than in the discussion part)

12. l.271 I do not fully understand yet the definition of $\mu$ " number of neighbours a point needs to be considered a core point, and therefore generate a new cluster". Does it refer to a maximum size of a cluster, beyond which the remaining gridpoints are considered as another cluster? And l.272 why does this value have to be larger than the number of dimensions plus one?

13. l.274 Please provide some reference about the tendency of ERA-5 to smooth extremes.

14. I can see that l.276 and 277 are some background to motivate the choice $\mu = 10$, but the link is not clear to me yet. l.279 "from at least 1 to 10 hours", is 10 hours a maximum duration? the "area between 5200 and 400km$^2$" is a duration of 1 h associated with maximum 5200 km, and 10h with maximum 400km$^2$ ? Are those value chosen to capture the convective events only, described in the previous sentences?

15. l. 285 about the "knee in the plot", maybe give some more details in Supplement 1 about how is exactly defined the knee. Is it the location of a specific value of the gradient?

16. l.295 it should be Fig. 6.

17. Why does the $10 - NN$ neighbourhood include $n_{max} = 44$ points?

18. Fig 7. This figure is a very good added value for understanding the definition of compound events you use. I guess the blue dashed lines mark out the compound event defined in the end. As it is present in the schematic of temporal overlap, maybe you can add it in the one of spatial overlap. And maybe use another color, so that it is more visible in a) b) c) and d).

19. l.331 Could you comment about the relevance of the choice AND for spatial and OR for temporal, in terms of impact?

20. l.345 Can you justify the choice of maximum precipitation as intensity attribute, and not accumulated precipitation over the period of the event?

21. l.371 "This confrontation highlights not only our methodology's capabilities, but also the ability of the ERA5 reanalysis to detect different types of extreme events in Great Britain": this is true if you find an agreement between the two datasets. In case of disagreement, how can you identify the source of it, between your methodology's capabilities and ERA5 performance?

22. l.497 "Among these 147 events, 64 (43.5%) have one corresponding cluster" maybe add "exactly" "Among these 147 events, 64 (43.5%) have *exactly* one corresponding cluster"

23. l.500 The reader can be confused between the origins of the hazards (the catalogue of observations and the output of your method), you can be a bit more specific, e.g. "109 are identified *by our algorithm* as compound hazard events".

24. l.515 I would recommend to use the variables you introduces in table 1, to be more specific: "sorted $p_a$ or $w$ at each gridpoints" instead of "the sorted sample"

25. In the caption of figure 13: describe in the corresponding order "Spatial-quantile plot for 18,086 precipitation, 6190 wind, and 4555 compounds cluster"

26. l.547 The spatial footprint is defined by the percentage of gridpoints concerned. However, the "real" size of the gridpoints depends on the latitude. Can this have an impact when comparing hazards in different regions of the studied area, in Figure 13?

27. l.560 I do not fully understand the end of the paragraph, could you reformulate your point?

28. l.565 maybe say that the event 136 is the one presented in figure 12b.

29. Can you comment on the 10 events present in the catalogue that the algorithm did not detect?

30. Briefly discuss about the identification of compound hazards in the context of climate change.

31. There are lots of results in appendix. You could maybe summarized them somewhere in the article itself?

32. l.637 *complementary*

33. Figure A1 why is the proportion of CE strictly increasing with the footprint area, and not with the duration? Maybe you can interpret a bit further Fig A1.

34. l.689 $LMF$ is interesting to study in this context. I would repeat the argument of the colorbar choice here: it should differentiate $LMF$ greater and lower than one.

35. l.706 Can you interpret the seasonality in terms of weather regimes?

36. l.713 Fig. A5?

37. l.731 It is the empirical probability, right?

38. Fig.A6, consider adding on the plots the cell of reference.

**References**

Kholodovsky, V. and Liang, X.-Z. (2021). A generalized spatio-temporal threshold clustering method for identification of extreme event patterns. *Advances in Statistical Climatology, Meteorology and Oceanography*, 7(1):35–52.

---

## Author Comment (AC1)

**"A Methodology for the Spatiotemporal Identification of Compound Hazards: Wind and Precipitation Extremes in Great Britain (1979–2019)" by Aloïs Tilloy, Bruce D Malamud and Amélie Joly-Laugel (discussion started 6 July 2021)**
**Reply to Referee # 1**

We thank anonymous referee # 1 for their time in reading and commenting critically on our manuscript ESD-2021-52 "A Methodology for the Spatiotemporal Identification of Compound Hazards: Wind and Precipitation Extremes in Great Britain (1979–2019)". We believe that the changes to our manuscript based on the reviewer comments will significantly improve it, and we hope this will be of use to the broader natural hazards' community. Below we respond in detail to the reviewer, first outlining the comments, and then our reply.

Lines numbers referred to by the reviewer are from our original manuscript submitted 6 July 2021, and line numbers referred to in our replies are from our revised manuscript.

We also attach to this reply a track change document of the changes made to date, with green indicating items that have been 'moved' from one section to another section.

**Authors Reply to Anonymous Referee #1**

**[R1 Overview]** "The authors present a method to identify compound hazards, in both space and time. The description of the method is very detailed and precise. A large catalogue of observed wind-precipitation hazards is additionally provided. It is used to assess the hazards identified by the algorithm applied on wind and precipitation data from the reanalysis dataset ERA-5. The appendix also contains several results that would deserve to be interpreted more in detail, and summarized in the article itself. The hit rate between events from the catalogue and events identified by the algorithm is quite high. However, it would be nice to see a discussion on the events that were not captured. This study is relevant and well described, this article deserves publication after minor revisions."

[Reply to General Comments] We thank the Anonymous Reviewer 1 (R1) for the positive feedback provided, and suggestion of (i) further interpretation in the text itself of our extensive appendix (thank you for looking at this in detail, as often reviewer pass over the appendices), (ii) and additional discussion on those events from the catalogue not identified by the algorithm events. Based on these comments, we have revised the manuscript. We have done the following:

- We have added a description of the appendix in the discussion (see also R1-31)
- As suggested by the reviewer, the colour scales of Figure 5, 11a, A2 and A3 have been modified along with the legend  and are now sequential colour maps, with homogeneous brightness gradient.
- The quality of ERA5 product for wind and precipitation extreme is now extensively discussed with new references. The attribution of missed events to the clustering method or ERA5 is also discussed in Section 6 of the manuscript.

Below, we provide detailed replies to the R1 specific comments (Author Replies—AR).

**[R1 Specific comments 1 to 38]**

[R1–1] l. 8 "Compound hazards are two different natural hazards". Actually two or more, as written in conclusion l.629
[Reply to R1–1] Agreed. The first sentence of the abstract has been changed accordingly.

[R1–2] "Even if you are not using it, consider mentioning methods of threshold selection, in the context of spatiotemporal clustering (e.g. Kholodovsky and Liang, 2021)"
[Reply to R1–2] We were not aware of this article, it is very relevant for this article, thank you. The reference is now discussed in revised manuscript (l.106 p.5).

[R1–3] "l.50 space missing between period and 1979-2019."
[Reply to R1–3] Change done

[R1–4] "l.56 the paragraph is a repetition of the previous one."

[Reply to R1–4] Change done

[R1–5] "l.67 repetition e.g."
[Reply to R1–5] Change done

[R1–6] "Maybe don't introduce acronyms/variables if you don't use them more than twice (CHE l.87, x and y l.90)"
[Reply to R1–6] Agreed. The acronyms have been removed.

[R1–7] "l.171 You mention the literature regarding the quality assessments of ERA-5 precipitation, but not on ERA-5 wind. Even if there is none, it can be worth mentioning it."
[Reply to R1–7] Agreed. The following three recent references are now mentioned and their findings are now discussed (Minola et al., 2020; Molina et al., 2021; Zscheischler et al., 2021) (l.169-172, p.8).

[R1–8] "Sometimes the word "footprint" refers to spatial footprint only, sometimes spatio-temporal footprint. I would recommend being consistent, and maybe always write "spatial footprint" when it is only about spatial considerations."
[Reply to R1–8] We agree this can be confusing. Thorough the manuscript, the word footprint refers to the spatial footprint only, even if this footprint is calculated over the aggregated duration of the compound hazards event. To make this less confusing, the words spatial and spatiotemporal in front of footprint have been removed and the definition of "footprint" in the study is given l.33: "*Here, the spatial scale (the 'footprint') refers to the area over which the hazard occurs*"

[R1–9] "It should be \Fig- 5."
[Reply to R1–9] Changes done (l.225)

[R1–10] A general remark about the figures: when plotting continuous variables, instead of qualitative color maps I would highly recommend using sequential color maps, with homogeneous brightness gradient. Consider changing color maps in figures 5., 11a., A2, A3 and A6.
[Reply to R1–10] Thank you for this comment. We did use sequential colours in the maps but failed to reflect this in the legend (thank you spotting this). We however acknowledge that the colour scales used were not optimal as they contained too many colours. After consultation of good practices, we changed the colour scales for the following figures:
- Figures 5 and A2, A3: Colour scales have been changed to sequential colours with sequential legends
- Figures 11 : The colour scale of (a) has been modified and the legend has been changed appropriately to also be sequential.

Finally, Figure A6 was left as it is as we think the colour gradient used brings out the gradation of patterns for spatial dependence.

[R1–11] "l.259 Please motivate a bit more the choice of r = a=b (earlier or/and in more details than in the discussion part)"
[Reply to R1–11] The action of clustering extreme values in time and space implies a weight in both dimensions. The main driver behind setting the parameter *r* is to make the spatiotemporal cube unitless and detect continuous clusters in time and space. Furthermore, the sensitivity analysis shows

that this parameter has a limited influence on the number of clusters detected. Sentences developing these points have now been added in Section 4.2.1 (l.257-263 p.13).

[R1–12] "l.271 I do not fully understand yet the definition of the number of neighbours a point needs to be considered a core point, and therefore generate a new cluster. Does it refer to a maximum size of a cluster, beyond which the remaining gridpoints are considered as another cluster? And l.272 [Section 4.2.2] why does this value have to be larger than the number of dimensions plus one?"
[Reply to R1–12] A detailed definition of core points is provided in Supplement 1. We now highlight this at the end of Section 4.2.2 and state at the beginning of Section 4.2.2 that the value of the density threshold must be larger than the number of dimensions plus one because it a requirement of the DBSCAN algorithm method (Ester, 1996).

[R1–13] "l.274 Please provide some reference about the tendency of ERA-5 to smooth extremes."
[Reply to R1–13] References for precipitation and wind extreme underestimation have now been provided in Section 4.2.2.

[R1–14] "I can see that l.276 and 277 are some background to motivate the choice x = 10, but the link is not clear to me yet. l.279 "from at least 1 to 10 hours", is 10 hours a maximum duration? the area between 5200 and 400km2" is a duration of 1 h associated with maximum 5200 km, and 10h with maximum 400km2 ? Are those value chosen to capture the convective events only, described in the previous sentences?"
[Reply to R1–14] We recognise that this section is probably a bit confusing. It means that the minimum spatiotemporal extent of a cluster is comprised between 1h-5200 km2 and 10h-400km2. The text in Section 4.2.2 has been modified to improve clarity.

[R1–15] "l. 285 about the "knee in the plot", maybe give some more details in Supplement 1 about how is exactly defined the knee. Is it the location of a specific value of the gradient?"
[Reply to R1–15] We did not use the gradient to identify the knee of the plot. The automatic detection is based on the computation of the distance between a straight line (green line in Figure 1) connecting the two extremities of the curve representing the sorted Euclidean distance to the 10th nearest neighbour (10-NN), (black dots in Figure 1). The knee is identified as the coordinate where the Euclidean distance between the green line and the 10-NN curve is the largest . This largest distance is represented by red dashed line in Figure 1 and its extremities are represented with red dots. The neighbour parameter $\varepsilon$ for a particular set of points is then the y-coordinate of the point on the 10-NN curve (red point on the 10-NN  curve in Figure 1) with the largest distance to straight line connecting the two extremities of the 10-NN curve. We have now added this previous sentence to Supplement 1 (p.3).

[Figure]

*Figure 1: Illustration of the knee automatic detection method*

[R1–16] "l.295 it should be Fig. 6."
[Reply to R1–16] Change done

[R1–16] "Why does the 10 -NN neighbourhood include nmax = 44 points?"
[Reply to R1–17] It is due to the values of the different parameters, this is illustrated in supplement figure 1.3 (we now mention this in the text l.300 p.15).

[R1–18] "Fig 7. This figure is a very good added value for understanding the definition of compound events you use. I guess the blue dashed lines mark out the compound event defined in the end. As it is present in the schematic of temporal overlap, maybe you can add it in the one of spatial overlap. And maybe use another color, so that it is more visible in a) b) c) and d)."
[Reply to R1–18] We have added a dashed line for spatial overlap and changed the colour to black to make it more visible.

[R1–19] "l.331 Could you comment about the relevance of the choice AND for spatial and OR for temporal, in terms of impact?"
[Reply to R1–19] We wanted our definition to be relevant in term of impact and this was a driver of our choice. A justification for the temporal scale is now provided [ it is p.16 For the spatial scale, we chose the AND to emphasize the compounding effect of triggering a potential impact. A sentence commenting on the relevance of the choice made in the article in terms of impact was added in the paragraph following Fig. 7 (l.335 p.17). A sentence opening on other potential definitions was also added in the discussion (l.620, p31).

[R1–20] "l.345 Can you justify the choice of maximum precipitation as intensity attribute, and not accumulated precipitation over the period of the event?"
[Reply to R1–20] The intensity attribute selected is the maximum accumulated precipitation over the period of the cluster. It represents a local maximum of accumulated precipitation for a cluster. It

echoes the peak wind gust which is also a local maximum over the period of the cluster. A clarifying sentence has been added (l.360 p.18).

[R1–21] "l.371 "This confrontation highlights not only our methodology's capabilities, but also the ability of the ERA5 reanalysis to detect different types of extreme events in Great Britain": this is true if you find an agreement between the two datasets. In case of disagreement, how can you identify the source of it, between your methodology's capabilities and ERA5 performance?"
[Reply to R1–21] That is an excellent question. The performance of the method developed here is assessed using a catalogue of major events built using different observational datasets which are not related to ERA5. This approach makes the disentangling of the influence of the clustering method (SI-CH) and the data (ERA5) difficult.  A way to do identify the source of the performance would be to apply the clustering method to a different dataset, ideally observational (e.g. CMORPH for precipitation). Additionally, The hit rate observed in this study for extreme precipitation is higher than the one obtained by to Rivoire et al. (2021). This could be suggest that the clustering method used in the present study improves the ability to identify extreme events with ERA5. However, differences in spatial and temporal resolution of the extremes and of the reference dataset make the comparison between the two studies complicated. This paragraph has been added to the discussion (l.628-635 p.31-32).

[R1–22] l.497 Among these 147 events, 64 (43.5%) have one corresponding cluster" maybe add exactly" Among these 147 events, 64 (43.5%) have exactly one corresponding cluster"
[Reply to R1–22] Change done

[R1–23] "l.500 The reader can be confused between the origins of the hazards (the catalogue of observations and the output of your method), you can be a bit more specific, e.g. "109 are identified by our algorithm as compound hazard events"."
[Reply to R1–23] Just above Fig. 11, the sentence you are referring to has now been clarified and another sentence has been added.

[R1–24] "l.515 I would recommend to use the variables you introduces in table 1, to be more specific: sorted pa or w at each gridpoints" instead of "the sorted sample""
[Reply to R1–24] In the sentences after Eqs. (3) and (4) (l.532 & l.535), we have now introduced the variables and improved clarity. Note that the total "sample" is the total amount of precipitation accumulation and wind peak values from all clusters.

[R1–25] "In the caption of figure 13: describe in the corresponding order "Spatial-quantile plot for 18,086 precipitation, 6190 wind, and 4555 compounds cluster""
[Reply to R1–25] Change done

[R1–26] "l.547 The spatial footprint is defined by the percentage of gridpoints concerned. However, the "real" size of the gridpoints depends on the latitude. Can this have an impact when comparing hazards in different regions of the studied area, in Figure 13?"
[Reply to R1–26] Yes, this can have an impact. As mentioned in Section 3, the "width" of a grid cell varies from 14.3 km to 18.6 km from north to south. This means that the cell real area is between 398 $km^2$ and 517 $km^2$ (change of approx. 30%). The approximations made here makes event occurring in

the northern part larger than they really are but we do not believe that the conclusions of the study and the shape of the curves in Fig. 13 are deeply impacted. Such approximation would be more impactful if applied on a wider area (e.g. pan European) and would probably require a modification of the grid. A sentence has been added after Fig. 13 (l.566-568 p.30).

[R1–27] "l.560 I do not fully understand the end of the paragraph, could you reformulate your point?"
[Reply to R1–27]. The paragraph (l.577-588 p.30) have been edited and reformulated. The point was to highlight the fact that one event in the catalogue can match more than one cluster from the database.

[R1–28] "l.565 maybe say that the event 136 is the one presented in figure 12b."
[Reply to R1–28] For more clarity, a new example is provided with event 62 which is associate with one large and intense cluster and numerous smaller clusters. Clusters matched with event 62 are now displayed in darkgreen in Figure 13a.

[R1–29] "Can you comment on the 10 events present in the catalogue that the algorithm did not detect?"
[Reply to R1–29] The 10 of 157 events present in the catalogue but not detected by the algorithm are heterogeneous with no clear seasonal pattern. Among these ten events, six do have temporally corresponding clusters, where clusters occur the same day as those detected by the algorithm but in other NUTS2 regions. These events are small or medium scale events (8 of the 10 reported in less than 3 NUTS2 regions). Finally, 7 out of 10 events are extreme precipitation events. The absence of clusters associated to some events means that there was not a sufficient amount of extreme values of wind/precipitation in the NUTS2 region where the significant event occur to trigger the creation of a cluster in that area. This could be due to the high value of the threshold for extreme values (q=0.99). Another explanation is that ERA5 could not reproduce these events, as the dataset can miss localized extreme, this is particularly true for precipitation (see relevant section). A paragraph based on this answer has been added below Fig. 11 (l.515-523 p.26).

[R1–30] "Briefly discuss about the identification of compound hazards in the context of climate change."
[Reply to R1–30] A sentence discussing the approach in the context of climate change has been added in the penultimate paragraph of the conclusion (l.675-678 p.33).

[R1–31] There are lot of results in appendix. You could maybe summarized them somewhere in the article itself?
[Reply to R1–31] We have now summarized the appendix main figures in the discussion.

[R1–32] l.637 complementary
[Reply to R1–32] Change done

[R1–33] "Figure A1 why is the proportion of CE strictly increasing with the footprint area, and not with the duration? Maybe you can interpret a bit further Fig A1."
[Reply to R1–33] We have added the following sentence to Appendix A: "A similar pattern is visible when the duration of the cluster increases (**Fig. A1b**), with a sharp increase of the proportion of

compound hazard clusters up to a duration of 30 h and a slow increase above that value. This could mean that above that duration, clusters belong to a physically homogeneous group that could be extra-tropical cyclones (as suggested by **Figs. A4** and **A5**).”

[R1–34] “l.689 LMF is interesting to study in this context. I would repeat the argument of the color bar choice here: it should differentiate LMF greater and lower than one.”
[Reply to R1–34] In Fig. A3, LMF is always greater than 1 with a minimum value of approximately 3 and a maximum value of 10. However, thanks to this comment, the LMF map has been updated as the one in the original manuscript was an outdated version (clustering without guard area). The Colorbar has been modified.

[R1–35] “l.706 Can you interpret the seasonality in terms of weather regimes?”
[Reply to R1–35] We tried using Lamb Weather types (Lamb, 1972) to find patterns in the occurrence compound hazard clusters, however the results were not very convincing. It is however likely that more appropriate weather regimes classification exist for the UK. It could be an interesting analysis to be conducted by those experts (e.g., at the Met Office) working with weather regimes, and we would welcome liaising with them if they were interested.

[R1–36] l.713 Fig. A5?
[Reply to R1–36] Change done

[R1–37] l.731 It is the empirical probability, right?
[Reply to R1–37] Yes it is. It is now clarified in the text.

[R1–38] Fig.A6, consider adding on the plots the cell of reference.
[Reply to R1–38] This is a good remark. In Fig. A6, the cell of reference is the only purple cell (Ps=1) and therefor is highlighted in this way.

**References**

Kholodovsky, V. and Liang, X.-Z.: A generalized spatio-temporal threshold clustering method for identification of extreme event patterns. Adv. Stat. Clim. Meteorol. Oceanogr., 7(1), 35–52, 2021, https://doi.org/10.5194/ascmo-7-35-2021.

Minola, L., Zhang, F., Azorin-Molina, C., Pirooz, A. A. S., Flay, R. G. J., Hersbach, H. and Chen, D.: Near-surface mean and gust wind speeds in ERA5 across Sweden: towards an improved gust parametrization, Clim. Dyn., 55(3–4), 887–907, 2020.

Molina, M. O., Gutiérrez, C. and Sánchez, E.: Comparison of ERA5 surface wind speed climatologies over Europe with observations from the HadISD dataset, Int. J. Climatol., (October 2020), 1–15, 2021.

Zscheischler, J., Naveau, P., Martius, O., Engelke, S. and C. Raible, C.: Evaluating the dependence structure of compound precipitation and wind speed extremes, Earth Syst. Dyn., 12(1), 1–16, 2021.

---

## Author Response (AR1)

**"A Methodology for the Spatiotemporal Identification of Compound Hazards: Wind and Precipitation Extremes in Great Britain (1979–2019)" by Aloïs Tilloy, Bruce D Malamud and Amélie Joly-Laugel**
**Reply to Referees # 1 and #2**

We thank the editor and both anonymous referees #1 [R1] and #2 [R2] for their time in reading and commenting critically on our manuscript ESD-2021-52 "A Methodology for the Spatiotemporal Identification of Compound Hazards: Wind and Precipitation Extremes in Great Britain (1979–2019)". We believe that the changes to our manuscript based on both of the reviewer comments will significantly improve it, and we hope this will be of use to the broader natural hazards community. Below we respond in detail to each reviewer, first outlining the comments, and then our reply.

Lines numbers referred to by the reviewers are from our original manuscript submitted 6 July 2021, and line numbers referred to in our replies are from our revised manuscript.

We also attach to this reply a track change document of the changes made to date, with green indicating items that have been 'moved' from one section to another section.

**Authors Reply to Anonymous Referee #1**

**[R1 Overview]** "The authors present a method to identify compound hazards, in both space and time. The description of the method is very detailed and precise. A large catalogue of observed wind-precipitation hazards is additionally provided. It is used to assess the hazards identified by the algorithm applied on wind and precipitation data from the reanalysis dataset ERA-5. The appendix also contains several results that would deserve to be interpreted more in detail, and summarized in the article itself. The hit rate between events from the catalogue and events identified by the algorithm is quite high. However, it would be nice to see a discussion on the events that were not captured. This study is relevant and well described, this article deserves publication after minor revisions."

[Reply to General Comments] We thank the Anonymous Reviewer 1 (R1) for the positive feedback provided, and suggestion of (i) further interpretation in the text itself of our extensive appendix (thank you for looking at this in detail, as often reviewers pass over the appendices), (ii) and additional discussion on those events from the catalogue not identified by the algorithm events. Based on these comments, we have revised the manuscript. We have done the following:

- We have added a much more detailed description of the appendix A in the discussion lines 655 to 664 (see also our reply to R1-31)
- As suggested by the reviewer, the colour scales of Figures 5, 11a, A2 and A3 have been modified along with the legend and are now sequential colour maps, with homogeneous brightness gradients.
- The quality of ERA5 product for wind and precipitation extreme is now extensively discussed with new references. The attribution of missed events to the clustering method or ERA5 is also discussed in Section 6 of the manuscript.

Below, we provide detailed replies to the R1 specific comments (Author Replies—AR).

**[R1 Specific comments 1 to 38]**

[R1–1] l. 8 "Compound hazards are two different natural hazards". Actually two or more, as written in conclusion l.629
[Reply to R1–1] Agreed. The first sentence of the abstract has been changed to now read "Compound hazards refer to two or more different natural hazards occurring over the same time period and spatial area."

[R1–2] "Even if you are not using it, consider mentioning methods of threshold selection, in the context of spatiotemporal clustering (e.g. Kholodovsky and Liang, 2021)"
[Reply to R1–2] We were not aware of this article, it is very relevant for this article, thank you. The reference is now discussed in the revised manuscript (l.111 p.5).

[R1–3] "l.50 space missing between period and 1979-2019."
[Reply to R1–3] Change done

[R1–4] "l.56 the paragraph is a repetition of the previous one."

[Reply to R1–4] Change done

[R1–5] "l.67 repetition e.g."
[Reply to R1–5] Change done

[R1–6] "Maybe don't introduce acronyms/variables if you don't use them more than twice (CHE l.87, x and y l.90)"
[Reply to R1–6] Agreed. The acronyms have been removed.

[R1–7] "l.171 You mention the literature regarding the quality assessments of ERA-5 precipitation, but not on ERA-5 wind. Even if there is none, it can be worth mentioning it."
[Reply to R1–7] Agreed. The following three recent references are now mentioned and their findings are now discussed (Minola et al., 2020; Molina et al., 2021; Zscheischler et al., 2021) (l.175-177, p.8).

[R1–8] "Sometimes the word "footprint" refers to spatial footprint only, sometimes spatio-temporal footprint. I would recommend being consistent, and maybe always write "spatial footprint" when it is only about spatial considerations."
[Reply to R1–8] We agree this can be confusing. Thorough the manuscript, the word footprint refers to the spatial footprint only, even if this footprint is calculated over the aggregated duration of the compound hazards event. The definition of 'footprint' is given l.40: "*Here, the spatial scale (the 'footprint') refers to the area over which the hazard occurs*". To make this less confusing, the words spatial and spatiotemporal in front of footprint have been removed throughout the manuscript.

[R1–9] "It should be \Fig- 5."
[Reply to R1–9] Changes done (l.231)

[R1–10] A general remark about the figures: when plotting continuous variables, instead of qualitative color maps I would highly recommend using sequential color maps, with homogeneous brightness gradient. Consider changing color maps in figures 5., 11a., A2, A3 and A6.
[Reply to R1–10] Thank you for this comment. We did use sequential colours in the maps but failed to reflect this in the legend (thank you spotting this). We however acknowledge that the colour scales used were not optimal as they contained too many colours. After consultation of good practices, we changed the colour scales for the following figures:

- Figures 5 and A2, A3: Colour scales have been changed to sequential colours with sequential legends
- Figures 11: The colour scale of (a) has been modified and the legend has been changed appropriately to also be sequential.

Finally, Figure A6 was left as it is as we think the colour gradient used brings out the gradation of patterns for spatial dependence.

[R1–11] "l.259 Please motivate a bit more the choice of r = a=b (earlier or/and in more details than in the discussion part)"
[Reply to R1–11] The action of clustering extreme values in time and space implies a weight in both dimensions. The main driver behind setting the parameter *r* is to make the spatiotemporal cube unitless and detect continuous clusters in time and space. Furthermore, the sensitivity analysis shows that this parameter has a limited influence on the number of clusters detected. Sentences developing these points have now been added in Section 4.2.1, in the lines following Eq. (1) (l.270-276, p.13).

[R1–12] "l.271 I do not fully understand yet the definition of the number of neighbours a point needs to be considered a core point, and therefore generate a new cluster. Does it refer to a maximum size of a cluster, beyond which the remaining gridpoints are considered as another cluster? And l.272 [Section 4.2.2] why does this value have to be larger than the number of dimensions plus one?"
[Reply to R1–12] A detailed definition of core points is provided in Supplement 1. We now highlight this at the end of Section 4.2.2 and state at the beginning of Section 4.2.2 that the value of the density threshold must be larger than the number of dimensions plus one because it a requirement of the DBSCAN algorithm method (Ester, 1996).

[R1–13] "l.274 Please provide some reference about the tendency of ERA-5 to smooth extremes."
[Reply to R1–13] References for precipitation and wind extreme underestimation have now been provided in Section 4.2.2.

[R1–14] "I can see that l.276 and 277 are some background to motivate the choice x = 10, but the link is not clear to me yet. l.279 "from at least 1 to 10 hours", is 10 hours a maximum duration? the area between 5200 and 400km2" is a duration of 1 h associated with maximum 5200 km, and 10h with maximum 400km2 ? Are those value chosen to capture the convective events only, described in the previous sentences?"
[Reply to R1–14] We recognise that this section is probably a bit confusing. It means that the minimum spatiotemporal extent of a cluster is comprised between (1 h, 5200 $km^2$) and (10 h, 400$km^2$). The text in Section 4.2.2 has been modified to improve clarity.

[R1–15] "l. 285 about the "knee in the plot", maybe give some more details in Supplement 1 about how is exactly defined the knee. Is it the location of a specific value of the gradient?"
[Reply to R1–15] We did not use the gradient to identify the knee of the plot. The automatic detection is based on the computation of the distance between a straight line (green line in Figure 1) connecting the two extremities of the curve representing the sorted Euclidean distance to the 10th nearest neighbour (10-NN), (black dots in Figure 1 of this document). The knee is identified as the coordinate where the Euclidean distance between the green line and the 10-NN curve is the largest . This largest distance is represented by red dashed line in Figure 1 and its extremities are represented with red dots. We have now added this following sentence to Supplement 1 (p.3). "The neighbour parameter $\varepsilon$ for a particular set of points is then the y-coordinate of the point on the 10-NN curve (red point on the 10-NN curve in Figure 1) with the largest distance to straight line connecting the two extremities of the 10-NN curve."

[Figure]

*Figure 1: Illustration of the knee automatic detection method*

[R1–16] "l.295 it should be Fig. 6."
[Reply to R1–16] Change done

[R1–16] "Why does the 10 -NN neighbourhood include nmax = 44 points?"
[Reply to R1–17] It is due to the values of the different parameters, this is illustrated in supplement figure 1.3 (we now mention this in the text l.313, p.15).

[R1–18] "Fig 7. This figure is a very good added value for understanding the definition of compound events you use. I guess the blue dashed lines mark out the compound event defined in the end. As it is present in the schematic of temporal overlap, maybe you can add it in the one of spatial overlap. And maybe use another color, so that it is more visible in a) b) c) and d)."
[Reply to R1–18] We have added a dashed line in Figure 7 for spatial overlap and changed the colour to black to make it more visible.

[R1–19] "l.331 Could you comment about the relevance of the choice AND for spatial and OR for temporal, in terms of impact?"
[Reply to R1–19] We wanted our definition to be relevant in term of impact and this was a driver of our choice. A detailed justification for the temporal scale is now provided, see p.16, l. 323-329. For the spatial scale, we chose the AND to emphasize the compounding effect of triggering a potential impact. A sentence commenting on the relevance of the choice made in the article in terms of impact was added in the paragraph following Fig. 7 (l.349, p.17). A sentence on other potential definitions has also been added in the discussion (l.631, p31).

[R1–20] "l.345 Can you justify the choice of maximum precipitation as intensity attribute, and not accumulated precipitation over the period of the event?"
[Reply to R1–20] The intensity attribute selected is the maximum accumulated precipitation over the period of the cluster. It represents a local maximum of accumulated precipitation for a cluster. It echoes the peak wind gust which is also a local maximum over the period of the cluster. A clarifying sentence has been added (l.373-4 p.18) which reads as follows: "Intensity attributes for both precipitation and

wind gust, as given above, represent a local maximum within clusters and not an average or a sum over the cluster footprint."

[R1–21] "l.371 "This confrontation highlights not only our methodology's capabilities, but also the ability of the ERA5 reanalysis to detect different types of extreme events in Great Britain": this is true if you find an agreement between the two datasets. In case of disagreement, how can you identify the source of it, between your methodology's capabilities and ERA5 performance?"
[Reply to R1–21] That is an excellent question. We have added the following paragraph to the discussion (l.644-651 p.31-32). "The method's performance developed here is assessed using a catalogue of major events built using different observational datasets that are not related to ERA5. This approach makes the disentangling of the influence of the clustering method (SI-CH) and the data (ERA5) difficult. A way to do identify the source of the performance would be to apply the clustering method to a different dataset, ideally observational (e.g. CMORPH for precipitation). Additionally, the hit rate observed in this study for extreme precipitation is higher than the one obtained by to Rivoire et al. (2021). This suggests that the clustering method used in the present study improves the ability to identify extreme events with ERA5. However, differences in spatial and temporal resolution of the extremes and of the reference dataset complicates comparing the two studies."

[R1–22] l.497 Among these 147 events, 64 (43.5%) have one corresponding cluster" maybe add exactly" Among these 147 events, 64 (43.5%) have exactly one corresponding cluster"
[Reply to R1–22] Change done

[R1–23] "l.500 The reader can be confused between the origins of the hazards (the catalogue of observations and the output of your method), you can be a bit more specific, e.g. "109 are identified by our algorithm as compound hazard events"."
[Reply to R1–23] Just above Fig. 11, the sentence you are referring to has now been clarified and another sentence has been added.

[R1–24] "l.515 I would recommend to use the variables you introduces in table 1, to be more specific: sorted pa or w at each gridpoints" instead of "the sorted sample""
[Reply to R1–24] In the sentences after Eqs. (3) and (4), we have now introduced the variables and improved clarity. Note that the total "sample" is the total amount of precipitation accumulation and wind peak values from all clusters.

[R1–25] "In the caption of figure 13: describe in the corresponding order "Spatial-quantile plot for 18,086 precipitation, 6190 wind, and 4555 compounds cluster""
[Reply to R1–25] Change done

[R1–26] "l.547 The spatial footprint is defined by the percentage of gridpoints concerned. However, the "real" size of the gridpoints depends on the latitude. Can this have an impact when comparing hazards in different regions of the studied area, in Figure 13?"
[Reply to R1–26] Yes, this can have an impact. As mentioned in Section 3, the "width" of a grid cell varies from 14.3 km to 18.6 km from north to south. This means that the cell real area is between 398 $km^2$ and 517 $km^2$ (change of approx. 30%). The approximations made here makes event occurring in the northern part larger than they really are but we do not believe that the conclusions of the study and the shape of the curves in Fig. 13 are deeply impacted. Such approximation would be more impactful if applied on a wider area (e.g. pan European) and would probably require a modification of the grid. A sentence for clarification has been added after Fig. 13 (top of p.30), which reads as follows: "The more

the curve goes toward the top right corner of the plot, the more severe the cluster is (high intensity over a large footprint). The footprint is expressed as the number of cells, which have a varying spatial area depending on the latitude (see **Sect. 3**), with cell areas in our study region varying from 398 km$^2$ (in the north) to 517 km$^2$ (in the south), a change of approximately 30%."

[R1–27] "l.560 I do not fully understand the end of the paragraph, could you reformulate your point?" [Reply to R1–27]. The paragraph (top of p.30, see our reply to R1–26) have been edited and reformulated. The point was to highlight the fact that one event in the catalogue can match more than one cluster from the database.

[R1–28] "l.565 maybe say that the event 136 is the one presented in figure 12b." [Reply to R1–28] For more clarity, a new example is provided (see end of Section 5, and Figure 13 caption) with event 62 which is associate with one large and intense cluster and numerous smaller clusters. Clusters matched with event 62 are now displayed in dark green in Figure 13a.

[R1–29] "Can you comment on the 10 events present in the catalogue that the algorithm did not detect?" [Reply to R1–29] Yes, thank you. We have now added the following paragraph after Fig. 11 (l.531-539 p.26).
"The 10 of 157 events present in our Great Britain weather events catalogue but not detected by the DBSCAN algorithm are heterogeneous with no clear seasonal pattern. Among these ten events, six have temporally corresponding clusters, where clusters occur the same day as those detected by the algorithm, but occur in other NUTS1 regions. The ten events are small- or medium-scale (8 of the 10 reported events occur in one or two NUTS1 regions) and 7 out of 10 events are extreme precipitation. The absence of clusters associated with some events means that there are not a sufficient number of extreme values of wind/precipitation in the NUTS1 region where the significant event occurs to trigger the creation of a cluster in that area. This could be due to the high value of the threshold for extreme values (q = 0.99). Another explanation is that ERA5 could not reproduce these events, as the dataset can miss localized extremes, particularly for precipitation (see Section 3)."

[R1–30] "Briefly discuss about the identification of compound hazards in the context of climate change."
[Reply to R1–30] We have now added the following to the penultipate paragraph of the conclusion (p.33). "The ability to analyse consistently the spatial and temporal attributes of climate related compound hazards is particularly relevant in the context of climate change as the intensity, frequency and spatiotemporal scales of single and compound hazards/are expected to change in the future due to human influences (Aghakouchak et al., 2020; Vogel et al., 2020; Spinoni et al., 2021)."

[R1–31] There are lot of results in appendix. You could maybe summarized them somewhere in the article itself?
[Reply to R1–31] We have now provided a summary of the appendix main figures in the discussion (lines 655 to 664).

[R1–32] l.637 complementary
[Reply to R1–32] Change done

[R1–33] "Figure A1 why is the proportion of CE strictly increasing with the footprint area, and not with the duration? Maybe you can interpret a bit further Fig A1."

[Reply to R1–33] We have added the following sentence to Appendix A: "A similar pattern is visible when the duration of the cluster increases (**Fig. A1b**), with a sharp increase of the proportion of compound hazard clusters up to a duration of 30 h and a slow increase above that value. This could mean that above that duration, clusters belong to a physically homogeneous group that could be extra-tropical cyclones (as suggested by **Figs. A4** and **A5**)."

[R1–34] "l.689 LMF is interesting to study in this context. I would repeat the argument of the color bar choice here: it should differentiate LMF greater and lower than one."
[Reply to R1–34] In Fig. A3, LMF is always greater than 1 with a minimum value of approximately 3 and a maximum value of 10. However, thanks to this comment, the LMF map has been updated as the one in the original manuscript was an outdated version (clustering without guard area). The Colourbar has been modified.

[R1–35] "l.706 Can you interpret the seasonality in terms of weather regimes?"
[Reply to R1–35] We tried using Lamb Weather types (Lamb, 1972) to find patterns in the occurrence compound hazard clusters, however the results were not very convincing. It is however likely that more appropriate weather regimes classification exists for the UK. It could be an interesting analysis to be conducted by those experts (e.g., at the Met Office) working with weather regimes, and we would welcome liaising with them if they were interested.

[R1–36] l.713 Fig. A5?
[Reply to R1–36] Change done

[R1–37] l.731 It is the empirical probability, right?
[Reply to R1–37] Yes it is. It is now clarified in the text.

[R1–38] Fig.A6, consider adding on the plots the cell of reference.
[Reply to R1–38] In Fig. A6, the cell of reference is the only purple cell (Ps=1) and therefore is highlighted in this way.

**Reply to Anonymous Referee # 2 [R2]**

[R2–1] "This manuscripts provides a clear and effective contribution to discerning and evaluating compound natural hazards, basically taking into consideration spatiotemporal clustering procedures to detect and classify the aggregation of such hazards with well explicited metrics (though naturally there are a lot of nuances and details enriching the work). Overall, the work is very transparent, well explained and operationable, especially bearing into account the provided supplements with data and code that can be aptly worked on in an understandable manner. As such, this provides a valuable contribution not only at scholarly level but also in operational services. This being said, and having seen the previous reviewer report, I will not repeat what is already there and to which I naturally concur. My minor remarks thus come down to the following aspects."

[Reply to R2–1] We thank the reviewer for their positive comments and providing four minor remarks to which we reply to below. We added to the manuscript some precisions about the method used to cluster extreme events and included extra references associated with identification of compound extremes, clustering and the physical interpretation of the results.

[R2–2] "To those who might wonder, why use the metrics and assumptions sustaining the methodological development and deployment in the manuscript, rather than other alternative ways to detect and potentially attribute the diagnostics made in the paper?"

[Reply to R2–2] We thank the reviewer for this comment. We have now examined again the current literature, a bit more critically, to add to what we wrote previously. To the knowledge of the authors, there are relatively few studies which attempt to spatially and temporally cluster extreme meteorological (or other) variables and with a focus on compound hazards. Two examples of existing methodologies consider a space-time cube where extremes are clustered (Zscheischler et al., 2013; Vogel et al., 2020) and two separated dimensions with different clustering rules (Birant and Kut, 2007). We adopted the former approach and we used the spatiotemporal ratio (Section 4.2.1) to control the relationship between space and time.

We have now added the following to the end of Section 2 to highlight the specificity of the method used in this article and as exemplars of existing studies: "In spatiotemporal clustering, some approaches consider time and space as separate dimensions (e.g., Birant and Kut, 2007) with distinct clustering rules while other approaches consider a space-time cube (e.g., Zscheischler et al., 2013; Vogel et al., 2020). In this article we adopt the latter approach and use the spatiotemporal ratio (Sect. 4.2.1) to control the relationship between space and time. Among other factors, the characteristics of the data used influence the choice of the spatiotemporal clustering method. Here we use climate reanalysis data, which are gridded data. Our approach aims to cluster extreme occurrences of climate variables, similarly to Kholodovsky and Liang (2021)."

We have also by examining the literature, added (along with appropriate words) the following sources AghaKouchak et al. (2020) [l. 697], Catto and Dowdy (2021) [l.669], Zhang et al. (2021) [l.319, 801] and De Angeli et al. (2022) [l.52, l.96].

[R2–3] "Several methodologies for spatiotemporal compound event identification fall prey to the self-fulfilling prophecy of detecting what we want to detect through tuning the methodologies. Fortunately in this paper the procedure is sufficiently objective to minimise such risk. Could the authors elaborate in brief terms how their methodological choices fare better in this regard than the panoply of traditional process-blind statistical methods?"

[Reply to R2–3] Thank you for your comment on the method. We believe that the method we use in our manuscript for clustering occurrences of extreme meteorological variables to detect compound hazard clusters, blends an existing method with a new approach and methodology that differs from previous methods looking at compound hazards. This is highlighted in Section 4.3 of the revised

manuscript. The method developed here relies on the DBSCAN algorithm, which is not a statistical tool but rather an unsupervised machine learning tool. We therefore rely on the procedures of this method to detect as objectively as possible our hazard clusters. The neighbour parameter can be set with an automated approach; however, for the definition of the extreme threshold, density threshold and spatiotemporal ratio, the authors relied on physical assumptions about the minimum size of extreme events to be detected and damage relevant threshold as explained in the methodology section of the manuscript (Section 4). Precisions about where our method stands in the literature have been given at the beginning of Section 2. A new sentence has now been added at the beginning of section 4 (l.205 p.9), summarising the nature of the assumptions taken in the clustering procedure.

[R2–4] "In keeping the interdisciplinary inter-domain philosophy of the ESD journal in mind, could the authors elaborate a little further on the physical interpretation of the results, namely linking to ocean-atmospheric and land-atmospheric aspects that might help explain the results?"... "…beefing up the geophysical reasoning to further help the more physically minded readers make further sense of the results and potentialities of the study, besides what was already made clear in that regard."

[Reply to R2–4] In Appendix A, the seasonality of single and compound hazard clusters is analysed. The link between weather systems and compound wind and precipitation extremes is discussed in Appendix A. Figure A3 highlights hotspots for compound wind and precipitation events. Several sentences linking the patterns of Fig. A3 to characteristics of the British climate are now added to our revised manuscript, which now reads as follows.

"**Figures A4**, **A5** and **A3** suggest that extratropical cyclones could influence compound wind and precipitation extremes, with a West-East pattern characteristic of the British island (Hulme and Barrow, 1997) and an increase in the frequency of compound events in the extended winter. The influence of cyclonic weather systems coming from the Atlantic on precipitation and wind extremes in Great Britain has been highlighted in previous work (Hawcroft *et al.*, 2012; Dowdy and Catto, 2017). However, this does not mean that every compound hazard cluster occurring during the extended winter is an extratropical cyclone but suggests that such weather systems are drivers of compound wind–precipitation extreme clusters**.** Other weather systems such as convective storms can also lead to compound wind and precipitation extremes (Zhang et al., 2021)."

We also agree that a deeper dive in seasonal patterns with a link to weather regimes (as suggested by reviewer 1) would allow us to extend the physical interpretations of the results, but do not delve into this deeply in this current manuscript.

[R2–5] "Overall, this is a solid contribution, clearly one of the rare occasions in which the preprint itself would already be a worthy final paper. By addressing the concerns of the other reviewer and minor ones remaining here, this review report intends essentially to slightly raise the bar of the work from very good to excellent. … Again, very good work, a short notch away from excellence."

[Reply to R2–5] Thank you for these comments.

**References Cited in our Reply**

AghaKouchak, A., Chiang, F., Huning, L. S., Love, C. A., Mallakpour, I., Mazdiyasni, O., Moftakhari, H., Papalexiou, S. M., Ragno, E. and Sadegh, M.: Climate Extremes and Compound Hazards in a Warming World, Annu. Rev. Earth Planet. Sci., 48, 519–548, https://doi.org/10.1146/annurev-earth-071719-055228, 2020.

Catto, J. L. and Dowdy, A.: Understanding compound hazards from a weather system perspective, Weather Clim. Extrem., 32(May 2020), 100313, https://doi.org/10.1016/j.wace.2021.100313, 2021.

De Angeli, S., Malamud, B.D., Rossi, L., Taylor, F.E., Trasforini, E. and Rudari, R.: A multi-hazard framework for spatial-temporal impact analysis, Intl. J. Disast. Risk Re., 102829, https://doi.org/10.1016/j.ijdrr.2022.102829, 2022. [In-Press]

Kholodovsky, V. and Liang, X.-Z.: A generalized spatio-temporal threshold clustering method for identification of extreme event patterns. Adv. Stat. Clim. Meteorol. Oceanogr., 7(1), 35–52, 2021, https://doi.org/10.5194/ascmo-7-35-2021.

Minola, L., Zhang, F., Azorin-Molina, C., Pirooz, A. A. S., Flay, R. G. J., Hersbach, H. and Chen, D.: Near-surface mean and gust wind speeds in ERA5 across Sweden: towards an improved gust parametrization, Clim. Dyn., 55(3–4), 887–907, 2020.

Molina, M. O., Gutiérrez, C. and Sánchez, E.: Comparison of ERA5 surface wind speed climatologies over Europe with observations from the HadISD dataset, Int. J. Climatol., (October 2020), 1–15, 2021.

Zhang, Y., Sun, X. and Chen, C.: Characteristics of concurrent precipitation and wind speed extremes in China, Weather Clim. Extrem., 32, 100322, https://doi.org/10.1016/j.wace.2021.100322, 2021.

Zscheischler, J., Naveau, P., Martius, O., Engelke, S. and C. Raible, C.: Evaluating the dependence structure of compound precipitation and wind speed extremes, Earth Syst. Dyn., 12(1), 1–16, 2021.